# Maternal 25-Hydroxycholecalciferol Supplementation Dynamically Altered Milk Fatty Acid and Amino Acid Profiles and Improves Sow Performance

**DOI:** 10.3390/ani15213160

**Published:** 2025-10-30

**Authors:** Liang Wang, Haitong Wang, Peipei Wen, Yikai Fan, Xiaoli Ren, Yongqing Li, Chu Chu, Li Liu, Juncheng Huang, Bo Hu, Huaiyong Zhang, Shujun Zhang

**Affiliations:** 1Key Laboratory of Agricultural Animal Genetics, Breeding and Reproduction, Education Ministry of China, Huazhong Agricultural University, Wuhan 430070, China; panda3785646@webmail.hzau.edu.cn (L.W.); htw0411@webmail.hzau.edu.cn (H.W.); wenpeipei@webmail.hzau.edu.cn (P.W.); fanyk@nercita.org.cn (Y.F.); renxl1990@163.com (X.R.); liyongqing@webmail.hzau.edu.cn (Y.L.); chu1999@webmail.hzau.edu.cn (C.C.); 2Shandong Asia Pacific Chinwhiz Group Limited Company, Qingdao 266061, China; 3Information Technology Research Centre, Beijing Academy of Agriculture and Forestry Sciences, Beijing 100097, China; 4Xinjiang Academy of Animal Science, No. 468, Ali Mountain Street, Shayibake District, Urumqi 830063, China; liuli17509991115@webmail.hzau.edu.cn (L.L.); 17509991115@163.com (J.H.); ceabinzhou@163.com (B.H.); 5Key Laboratory of Animal Biochemistry and Nutrition, Ministry of Agriculture, College of Animal Science and Technology, Henan Agricultural University, Zhengzhou 450046, China; huaiyongzhang@henau.edu.cn

**Keywords:** 25-hydroxycholecalciferol, sow performance, fatty acid profile, amino acids

## Abstract

**Simple Summary:**

Given the presence of vitamin D receptors in mammary glands and their more efficient absorbability, this study was performed to evaluate the effects of maternal 25-hydroxycholecalciferol (25-OH-D_3_) supplementation on sow performance, as well as the dynamic alterations of both the amino acid and fatty acid profiles in milk. Herein, the concentration of all amino acids in milk was higher in colostrum, following a dramatic drop. The rate of reduction for all amino acids was increased by dietary 25-OH-D_3_ supplementation. The addition of dietary 25-OH-D_3_ initially suppressed both saturated and unsaturated fatty acid levels from days 1 to 7. Subsequently, the 25-OH-D_3_ group exhibited elevated levels of specific fatty acids relative to the control during l4 to 21 d of lactation, especially oleic and linoleic acids and arachidonic acid. These findings support the hypothesis that maternal supplementation of 25-OH-D_3_ in gestation and lactation diets could alter milk composition, which might be linked to the improved body weight of piglets, with an effective maintenance of body condition in sows.

**Abstract:**

This study was performed to evaluate the effects of maternal 25-hydroxycholecalciferol (25-OH-D_3_) supplementation on sow performance, as well as the dynamic alterations of both the amino acid and fatty acid profiles in milk. On day 85 of gestation, twenty primiparous hybrid sows were allocated into two groups (10 sows/group) and fed a basal diet (3200 IU/kg vitamin D_3_) containing either 0 or 50 μg/kg 25-OH-D_3_ until weaning on d 21 of lactation. Milk was collected at 1, 3, 7, 14, and 21 d of lactation. The results showed that dietary 25-OH-D_3_ supplementation notably decreased the score of tear stain at 101 d of gestation when compared to the Ctrl group (*p* = 0.030). No significant difference was found in terms of the gestation day, litter size, and litter weight at birth, whereas maternal 25-OH-D_3_ intervention notably increased weaning weight and weight gain of the piglet (*p* < 0.05), while dietary 25-OH-D_3_ supplementation contributed to a 16.4% body gain during lactation. The concentration of all amino acids in milk was higher in colostrum, following a dramatic drop. The rate of reduction for all amino acids was increased by dietary 25-OH-D_3_ supplementation. The contents of saturated fatty acids and polyunsaturated fatty acids were increased and decreased linearly throughout lactation (both *p* < 0.05). Dietary 25-OH-D_3_ supplementation initially suppressed both saturated and unsaturated fatty acid levels from d 1 to 7, while prompting a recovery of specific fatty acids from d 14 to 21 of lactation, particularly oleic acid, linoleic acid, and arachidonic acid. These findings indicate that maternal 25-OH-D_3_ supplementation alters the pattern of milk fatty acid and amino acid composition, which may be associated with the observed improvement in piglet outcomes.

## 1. Introduction

The global pig industry faces an increasing demand for sustainable production and high-quality meat. For sow performance, productivity and profitability are primarily determined by litter size and weaning weights, which are greatly affected by maternal nutrition and milk composition [1]. Particularly in late gestation, piglets undergo a rapid development, requiring sufficient maternal nutrition to meet fetal demands [2]. Multiple studies have indicated that nutritional interventions during gestation can alter organ structure and influence prenatal and neonatal growth, as well as weight gains in newborn pigs [3]. Therefore, providing adequate nutrients or targeted interventions throughout late pregnancy and lactation is critically important.

25-hydroxycholecalciferol (25-OH-D_3_) is the major circulating metabolite of vitamin D, produced in the liver. It is three times more potent than standard vitamin D at raising vitamin D status in older adults [4]. The benefits of vitamin D extend beyond its well-known role in calcium (Ca) and phosphorus (P) homeostasis to include immune function and production. Studies have found that the levels of 25-OH-D_3_ in serum and follicular fluid were related to ovarian development [5,6]. Clinical data showed that women with reproductive dysfunction had decreased serum levels of 25-OH-D_3_ compared to normal individuals [5,7]. In the pig-breeding industry, the dietary supplementation of 25-OH-D_3_ at 50 or 200 µg/kg to sows had no significant effect on the number of piglets born alive, litter weight, or suckling piglet performance [8,9]. However, maternal 25-OH-D_3_ supplementation at a dosage of 50 μg/kg has been reported to increase weaning litter weight and total litter weight gain [10,11]. Potential factors contributing to these results include the total dosage of vitamin D_3_, differences in feeding management, the duration of the trial, and the parity of the sows. Maternal vitamin D insufficiency was found to be linked to decreased fertility and adverse pregnancy outcomes [12], which were also accompanied by a reduction in breast milk vitamin D content, offspring birth weight, and neonatal bone mineral content [13]. Supplementing sow diets with 50 μg/kg of 25-OH-D_3_ has been shown to significantly increase weaning litter weight, with reported gains of 10.0% [11], up to 19.7% [10]. These findings suggest that supplementing sows with 25-OH-D_3_ during gestation and lactation may ensure an adequate maternal vitamin D status, thereby improving sow performance and offspring development.

Suckling pig growth depends on the availability and composition of milk, as well as the efficient conversion of its nutrients into body weight, suggesting that the growth performance of the majority of the litter relies on the amount and quality of colostrum and milk [14]. For example, threonine was linked to the production of immunoglobulin G (IgG) and played important roles in neonatal survival and infection prevention [15]. The supplementation of leucine is critical for the intestinal development of suckling pig through enhancing muscle protein synthesis and gut maturation [16]. Some specific fatty acids were also observed to exert an indispensable role in the development of piglets, such as oleic and linoleic acids. The improvement in piglet growth and survivability associated with dietary oleic and linoleic acid supplementation is attributed to their role as preferred energy substrates [17], their anti-inflammatory properties [18], and their ability to enhance mucosal development [19]. Interestingly, the vitamin D receptor and 1α-hydroxylase, a key enzyme responsible for converting vitamin D into its active form (1,25-(OH)_2_-D_3_), are presented in mammary glands [20]. Evidence has confirmed that the expression of sterol regulatory element-binding protein 1 (SREBP1) and its downstream target enzymes that mediate fatty acid synthesis were involved in the activation of vitamin D receptors [21]. Dietary 25-OH-D_3_ intervention promoted the expressions of acetyl-CoA carboxylase (ACC) and fatty acid synthase (FAS) in the breast tissue of lactating sows [22]. These findings imply that maternal 25-OH-D_3_ intervention could modify the milk composition. A previous study has shown that maternal supplementation with 25-OH-D_3_ has been shown to increase the concentrations of protein and lactose in milk during lactation [10]. The elevated fat content was also noticed in both colostrum and mature milk on d 21 of lactation due to dietary manipulation with 25-OH-D_3_ [9,23]. Previous studies have confirmed that dietary 25-OH-D_3_ alters fatty acids in milk, the mammary gland, and bone marrow [22,23]. Nonetheless, the effects of maternal 25-OH-D_3_ inclusion on the milk composition of amino acids and fatty acids require further elaboration.

Given this background, the objectives of the current research were to 1) investigate the effects of maternal 25-OH-D_3_ supplementation during lactation on sow performance and litter growth and 2) define the dynamic alterations in milk amino acid and fatty acid profiles in response to 25-OH-D_3_ and the day of lactation. This study provides mechanistic insights to support the application of 25-OH-D_3_ in sow diets to improve maternal and neonatal health.

## 2. Materials and Methods

### 2.1. Animals, Diets, and Management

Using a randomized block design, a total of 20 primiparous Duroc × (Landrace × Large White) gestating sows were allocated into two groups (10 sows per group) at 85 d of gestation based on initial body weight and backfat thickness. The sows were fed a basal diet (3200 IU/kg vitamin D_3_) containing 25-OH-D_3_ at either 0 μg/kg (Ctrl group) or 50 μg/kg (corresponding to 2000 IU/kg vitamin D_3_), according to previous study [10]. Dietary treatments were performed from 85 d of pregnancy to d 21 of lactation. The basal diets for gestation and lactation were formulated according to the nutrient requirements of the National Research Council (NRC, 2012; Table 1). All vitamins were purchased from DSM Nutritional Products Ltd. (Shanghai, China). Sows were maintained in individual stalls (2.13 × 0.61 m) for the duration of gestation (d 85–110) and provided a gestation diet at a level of 2.70 kg per day, divided into two feedings. On d 110 of gestation, sows were moved to farrowing pens and offered a lactation diet ad libitum. Feed intake was recorded via electronic feeders, which measured refusals to calculate daily consumption. After 3 d, sows had ad libitum access to water and lactation feed (Table 1). The farrowing house temperature was maintained at a minimum of 20 °C, with supplemental heat provided to piglets via heat lamps. Routine management procedures followed standard protocols. 

### 2.2. Backfat Thickness and Sow Body Condition Score

Backfat thickness was measured by a single researcher on d 85 of pregnancy, delivery, and d 20 of lactation at the P2 site using a digital backfat meter (Renco Lean-Meter, Minneapolis, MN, USA) [24], while the point of measurement was marked on each sow to guarantee that exactly the same place was investigated during the subsequent measurements. In addition, backfat thickness was also measured using a caliper by gently placing it at the marked site on the back of sows, ensuring firm skin contact without excessive pressure. The caliper arms were positioned on either side of the back, centered over the measurement point, and the displayed value was recorded, referring to previous methods [25].

The body condition score was recorded on d 93, 101, and 108 of gestation through visual estimation and by assessing the ease of feeling the hipbone and backbone of the sow on a traditional 5-point scale, described, for example, by Li et al. (2012) [26], which is shown in Table 2.

### 2.3. Tear Stain Score

Following standardized protocols [27], a trained photographer captured bilateral ocular images for tear stain assessment on gestation d 93, 101, and 108 using a digital camera (Canon Inc., Tokyo, Japan). During imaging, a handler maintained proper sow positioning. Image analysis was performed using ImageJ software (Version 1.54r, NIH, Bethesda, MD, USA), with calibration achieved by scaling the iris diameter to 1 cm. Rust-brown periocular deposits were quantified, with the cumulative stain area (cm^2^) scored bilaterally according to Table 2. The mean score from both eyes of all animals was used for analysis.

### 2.4. Performance of Sows

Body weight (BW) of sows was obtained at breeding, d 110 of gestation, within 24 h post-farrowing, and weaning. Daily feed intake was recorded during gestation (from d 85 to 110) and lactation, and total consumption was calculated per sow. At parturition, the numbers of live-born, stillborn, mummified, and malformed piglets were registered to determine total born and live-born litter sizes. Litter birth weight was measured, and individual piglet birth weights were calculated by dividing total litter weight by the number of live-born piglets. After farrowing, litter size was moved between sows within each group to ensure the survival rate and weaning weight of piglets. The adjusted litter weight and size were used to calculate the individual and litter weight gain. During the lactation period (d 1–20), piglet count was recorded daily to calculate survivability rates.

### 2.5. Measurement of Fatty Acids in Milk

Approximately 10 mL milk samples were collected by manual expression from all functional mammary glands at 1, 3, 7, 14, and 21 d of lactation after of intramuscular administration of 1.5 mL oxytocin solution (10 IU/mL; Løvens Kemiske Fabrik, Ballerup, Denmark). Fatty acid profile in milk for an individual sow was quantified by gas–liquid chromatography. Herein, according to previous description [28], fatty acids in milk were extracted, saponified, and esterified, and 500 mg milk samples were mixed with 0.50 mL water, 2.00 mL methanol, and 1.00 mL chloroform. The mixture was vortexed for 1 min, followed by sequential addition of 1.00 mL water and 2.00 mL chloroform. After centrifuging at 1000× *g* for 10 min, the lower chloroform phase was collected and subjected to saponification with NaOH and trans-esterification with boron trifluoride–methanol. The fatty acid methyl esters were analyzed using capillary gas–liquid chromatography, following the method described by Lin and colleagues [29].

### 2.6. Analysis of Amino Acid in Milk

The amino acid composition of milk samples was determined through acid hydrolysis as previously described [30]. Prior to hydrolysis, the samples were dried using a Savant speed-vac system and further desiccated with potassium hydroxide pellets and phosphorus pentoxide. The dried samples were then mixed with 6 mol/L HCl containing 1% phenol and 6% thioglycolic acid. Hydrolysis was performed under vacuum in an inert nitrogen atmosphere to prevent oxidative degradation of amino acids. Then, the sample was dried under vacuum at 110 °C for 24 h and reconstituted in lithium citrate buffer. The amino acid composition was quantified using a Biotronik LC 6001 Amino Acid Analyzer (Biotronik, Maintal, Germany) with post-column ninhydrin detection, measuring absorbance at 570 nm (440 nm for proline).

### 2.7. Statistical Analysis

Data were analyzed using the GLIMMIX procedure in SAS (version 9.4, SAS Institute, Inc., Cary, NC, USA) and considered sow (litter) as the study unit. Results are expressed as means and their standard errors. The data obtained were analyzed by the Shapiro–Wilk and Levene’s test to assess normal distribution and homogeneity of variances. An independent *t*-test was used to compare significant differences between the diet with and without 25-OH-D_3_ groups on sow performance, fatty acids, and amino acids. A Bonferroni correction was applied to control for multiple comparisons. one-way analysis of variance (ANOVA) followed by Tukey’s test for multiple comparisons was performed to elucidate the effect of lactation day on the fatty acid and amino acid profiles. The statistical model applied was as follows:$$Y_{i}=\;\mu \;+\;D_{i}+\;{Ɛ}_{i}$$
where $Y_{i}\;$ is the response variable, μ is the overall mean, $D_{i}$ is the fixed effect of dietary 25-OH-D_3_ or day of lactation, and ${Ɛ}_{i}$ is the error term.

In addition, the performance of sows during lactation between Ctrl and 25-OH-D_3_ groups was compared using one-way analysis of covariance (ANCOVA). Adjusted litter size was included as a covariate in a single ANCOVA model.

The alterations in amino acid and fatty acid profiles in resonance to lactation stages were assessed using the general linear mixed procedure, quadratic and cubic polynomials, logistic, one-phase decay, and two-phase decay models based on the coefficient of determination (R) using SAS with the Gauss–Newton algorithm. The one-phase decay model and the general linear mixed procedure were finally selected as the optimized models for the effects of dietary supplementation at different lactation stages on amino acid and fatty acid profiles, respectively. The one-phase decay model is described by the following equation and illustrated in Figure 1.Y = (Yo − Plateau) ∗ exp(−K ∗ Day) + Plateau.

Yo is the Y value when Day is 1 d of lactation, Plateau is the Y value at infinite times, and K represents the rate of reduction.

Statistical significance was set at *p* < 0.05, while results with *p*-values between 0.05 and 0.10 were deemed to indicate a statistical trend.

## 3. Results

### 3.1. Tear Stain Score and Body Condition

As presented in Table 3, dietary 25-OH-D_3_ supplementation notably decreased the score of tear stain at 101 d of gestation when compared to the Ctrl group (*p* = 0.030). There was no effect of the treatment on any aspect of body condition score, backfat depth measurement, or loss of backfat during lactation (*p* > 0.05).

### 3.2. Sow Performance

The supplementation of 25-OH-D_3_ in this study did not obviously change the body weight during gestation and lactation (Table 4). No significant difference was also found in terms of the gestation day, in which the gestation day was 116.4 ± 0.7 and 116.2 ± 1.14 in the Ctrl and 25-OH-D_3_ groups, respectively. A supplementation of 25-OH-D_3_ in this gestation diet did not obviously alter (*p* > 0.05) the litter size, including the numbers of live-born, stillborn, mummy, deformity, and total born (Table 4). The litter weight at birth (not including deformity) and body weight of piglets at birth were also comparable between treatments. With regard to the feed intake, which was investigated from 85 to 110 d of gestation and lactation, there was no effect of the treatment (*p* > 0.05) on feed consumption between the treatment and inspection time (Figure 2).

As presented in Table 5, at the start of lactation, the litter size was standardized within each group. The adjusted number of live-born piglets was higher (*p* < 0.001) in the Ctrl group (12.9 ± 0.88) than in the 25-OH-D_3_ group (11.0 ± 0.67). Therefore, ANCOVA was performed with the adjusted litter size serving as a covariate in a single ANCOVA model. The analysis showed that the numbers of weaned piglets and survivability to weaning were similar between groups (Table 4). While initial body weights did not differ, weaning weights on d 21 were significantly heavier in the 25-OH-D_3_ group (6.16 ± 0.48 kg) than in the Ctrl group (5.50 ± 0.77 kg; *p* = 0.031). Consequently, body weight gain from d 1 to 21 was greater in the 25-OH-D_3_ group compared to the Ctrl group (*p* = 0.016), in which the dietary 25-OH-D_3_ supplementation contributed to a 16.4% body gain during lactation.

### 3.3. Effects of Lactation Stage and Dietary 25-OH-D_3_ on the Milk Amino Acid Profile

In response to the lactation day, the concentration of all amino acids in milk was higher in colostrum, following a dramatic reduction (Table 6 and Figure 3). According to a one-phase decay model, the rate of reduction (K) for all amino acids was increased by dietary 25-OH-D_3_ supplementation. The content in the plateau was increased in terms of glutamic acid, arginine, methionine, serine, valine, alanine, and leucine due to dietary 25-OH-D_3_ intervention, where it decreased the plateau levels of aspartic acid, lysine, histidine, threonine, proline, tyrosine, phenylalanine, and isoleucine (Table 6).

When compared to the Ctrl group, the diet containing 25-OH-D_3_ resulted in a decreased concentration of amino acids in milk, especially at d 21 of lactation (*p* < 0.05, Table 6 and Figure 3B). To be specific, the diet containing 25-OH-D_3_ significantly elevated the content of glutamic acid at 21 d of lactation and the levels of alanine at 14 d of lactation (both *p* < 0.05). Conversely, it remarkably reduced the contents of lysine and histidine at 3 d of lactation, as well as proline at 7 d of lactation. Furthermore, a decreasing trend was observed for lysine, proline, and phenylalanine at d 7, and for isoleucine at d 14 and 24 (Table 6 and Figure 3).

### 3.4. Influence of Lactation Day and Dietary 25-OH-D_3_ Inclusion on the Fatty Acid Profile of Milk

As shown in Table 7 and Figure 4, as far as the Ctrl group was concerned, accompanied by very little fluctuation in monounsaturated fatty acids (*p* = 0.280), the saturated fatty acids (r = 0.496, *p* = 0.001) and polyunsaturated fatty acids (r = −0.494, *p* = 0.001) were linearly increased and decreased with the lactation days, respectively, contributing to comparable total fatty acid content (*p* > 0.05) and linearly reducing the ratio of unsaturated to saturated fatty acids (r = −0.581, *p* < 0.001) in milk. Dietary 25-OH-D_3_ supplementation initially decreased the content of both saturated and unsaturated fatty acids from d 1 to 7 of lactation. Subsequently, the 25-OH-D_3_-fed group showed higher levels of saturated fatty acids at d 14, monounsaturated fatty acids at d 21, and polyunsaturated fatty acids at d 14 and 21 compared to the control group. Consequently, the overall fatty acid content increased from 14 to 21 d after an initial decline. Furthermore, the ratio of unsaturated to saturated fatty acids decreased between 7 and 14 d but increased again by d 21.

From the perspective of saturated fatty acids, the concentrations of decanoic, lauric, and myristic acids increased, while those of heptadecanoic and stearic acids decreased throughout lactation (Figure 5A–F). As presented in Figure 6, the colostrum from sows supplemented with 25-OH-D_3_ exhibited a significant (*p* < 0.05) decrease in the content of palmitic, heptadecanoic, and stearic acids compared to the Ctrl group. These reductions were maintained until d 3 of lactation, with the decrease in stearic acid extending to d 7. On d 21 of lactation, the diet supplemented with 25-OH-D_3_ significantly decreased (*p* < 0.05) the content of decanoic, lauric, myristic, and palmitic acids. In contrast, the same diet increased (*p* = 0.012) palmitic acid levels on d 14 and stearic acid levels on d 21 (Table 6).

The effect of dietary 25-OH-D_3_ on monounsaturated fatty acids was time- and acid-specific (Table 7 and Figure 6G–K). While the treatment decreased levels of 10-heptadecenoic acid, oleic acid, and 11-eicosenoic acid during the two weeks of lactation, it led to a subsequent increase (*p* < 0.05) in oleic acid and 11-eicosenoic acid by d 21. Furthermore, at 21 d of lactation, the content of myristelaidic acid and palmitoleic acid remained lower than in the Ctrl group (*p* < 0.05).

As far as polyunsaturated fatty acid is concerned, its results are presented in Table 7 and Figure 5 and Figure 6L–P. Dietary 25-OH-D_3_ intervention had a time-dependent effect on polyunsaturated fatty acids. The diet containing 25-OH-D_3_ significantly decreased (*p* < 0.05) the levels of linoleic acid, α-linolenic acid, 11.14-eicosadienoic acid, and arachidonic acid in colostrum, but the dietary inclusion of 25-OH-D_3_ increased (*p* < 0.05) them from 14 to 21 d of lactation. Additionally, the treatment contributed to a higher concentration of γ-linolenic acid at d 21 compared to the Ctrl group (*p* = 0.033).

## 4. Discussion

Researchers are increasingly interested in how maternal nutrient supplementation and milk composition together influence piglet performance from birth to weaning [1]. In the present study, the trial period concentrated on the gestation and lactation period, and the influence of maternal 25-OH-D_3_ supplementation on sow performance and milk compositions was evaluated using a sow–piglet model. Several studies have established that maternal 25-OH-D_3_ supplementation can improve sow performance and health, supporting its potential application in diets [31]. The outcomes from this study suggested that, although sow body condition was unaffected, the maternal supplementation of 50 μg/kg 25-OH-D_3_ increased the weaning weights of piglets, which probably was associated with the alterations in milk composition, specifically its fatty acid and amino acid profiles.

The rapid evolution of growth potential and management practices in commercial swine has redefined optimal nutrition strategies. This new paradigm involves tailoring vitamin levels to specific physiological stages to achieve benefits that extend far beyond the simple production of deficiency. It was pointed out that tear staining is associated with ear and tail damage in sows [32], and isolation and lack of enrichment led to higher tear stain [27]. In this study, the tear stain score, a measure of stress usually used in rodent models and more recently in pigs [33], was conducted and found that dietary 25-OH-D_3_ inclusion notably decreased the score of tear stain at d 101 of gestation. It is therefore possible that dietary 25-OH-D_3_ supplementation might alleviate sow stress throughout farrowing and lactation. However, more convincing evidence is required to provide support for this concept in the present study. During the lactation, to meet the energy and precursor demands of milk production, sows mobilize body fat and protein when dietary nutrient intake is insufficient [34]. This mobilization negatively impacts litter growth and subsequent reproductive performance [35]. As a primary indicator of energy reserves, backfat thickness is key for assessing the capacity of sows to sustain fetal growth, farrowing, and lactation. A previous study showed that a dietary supplementation of 50 μg/kg 25-OH-D_3_ failed to change the backfat loss of sows during lactation and may contribute to equal amounts of milk production [11]. A trend in increased backfat thickness was observed during gestation when feeding diets contained 25-OH-D_3_, whereas no significant difference was found in sows fed 25-OH-D_3_ during the wean-finishing period [36]. In the current study, the body condition score, backfat thickness during gestation, and loss of backfat during the lactation of sows did not differ due to dietary 25-OH-D_3_ treatment. The similar alterations in sow backfat thickness might be related to the comparable feed consumption. The existing evidence indicated that the change in backfat thickness of sows had a high correlation with body weight loss and feed intake [37]. According to the performance of sows, including body weight change and feed consumption, this study also indicated that sows fed the 25-OH-D_3_ diet maintained a balance between anabolism and catabolism, which may improve longevity.

In addition to the well-known regulation of Ca and P metabolism, a growing body of literature has shown that the levels of 25-OH-D_3_ in serum and follicular fluid are closely linked to ovarian development [5,6]. Studies have found that serum contents of 25-OH-D_3_ were remarkably lower in women with reproductive dysfunction compared to normal controls. Furthermore, increasing 25-OH-D_3_ levels demonstrated a notable positive correlation with both estradiol concentrations and fertilization rates [5,7]. Treating isolated porcine ovarian granulosa cells with 1,25-(OH)_2_-D_3_ significantly altered the transcription and translation of genes regulating progesterone biosynthesis [6]. Maternal vitamin D insufficiency is linked to decreased fertility and adverse pregnancy outcomes [12]. This condition is also accompanied by a reduction in breast milk vitamin D content, offspring birth weight, and neonatal bone mineral content [13]. It seems that maintaining an appropriate concentration of 25-OH-D_3_ could mediate the selection of superior sperm for fertilization [38]. In this study, sows supplemented with 25-OH-D_3_ did not change in terms of their litter size and weight, which align with other studies. For example, evidence from Long et al. (2024) found that 50 µg/kg 25-OH-D_3_ supplementation had no significant effects on the number of piglets alive, stillborn rate, and litter weight [8]. Of note, this supplementation was administered from d 85 of gestation, when embryo implantation had already occurred. Therefore, it is normal that there were no differences in the size and number of piglets born. Maternal supplementation of vitamin D_3_ (1500–6000 IU/kg) [39] or 200 µg/kg 25-OH-D_3_ [9] were also found to not affect litter size criteria or suckling pig performance. However, a study on gilts found that a diet containing 50 μg/kg 25-OH-D_3_ during gestation could increase litter size or promote the birth body weight of newborn piglets, which might be due to differences in the amount of total vitamin D_3_, feeding managements, trial period, and the parity of sows. More importantly, as a key indicator of sow reproductive performance and economic efficiency, weaning litter weight was notably increased by maternal 25-OH-D_3_ supplementation in this study. The accompanying rise in total litter weight gain indicates that dietary 25-OH-D_3_ improves the piglet growth rate. This result corroborates the literature, where 25-OH-D_3_ supplementation at a dosage of 50 μg/kg has been reported to increase weaning litter weight by 19.7% [10]. Supplementing sows with 50 μg/kg 25-OH-D_3_ increased litter weaning weight and total litter weight gain by 10.0% and 13.3%, respectively [11]. The improved growth rate for piglets might derive from the interplay between 25-OH-D_3_ and the insulin-like growth factor (IGF)-1 system [36]. IGF-1, produced by the liver in response to growth hormones, works together with growth hormones (GHs) to promote whole-body growth and development. Studies have shown that maternal 25-OH-D_3_ manipulation notably increases the concentration of IGF-1 in the serum of sows and piglets at weaning [22]. The piglets from 25-OH-D_3_-fed sows exhibited a higher concentration of serum GH and IGF-1 in serum [22]. The positive relationship between serum 25-OH-D_3_ and IGF-I levels has been confirmed in the previous study [40]. Based on this external evidence, it is possible that the growth enhancement observed in piglets from sows supplemented with 25-OH-D_3_ might be due to the activation of the IGF-1 axis.

It is well established that the growth performance of a litter mainly relies on the amount and quality of colostrum and milk [14]. Although milk yield was not directly measured in this study, sow feed intake and backfat loss might be established indicators of milk production, as nutrient intake is partitioned between milk synthesis and maternal tissue retention [25]. Therefore, the undifferentiated feed intake and backfat loss of sows between Ctrl sows and 25-OH-D_3_ sows during lactation may, to some extent, reflect a similar amount of milk production, implying that the milk quality might be related to the enhancements in litter and piglet weight gain during lactation in the current study. The growth of suckling pigs was determined by both the availability and composition of milk and the efficient conversion of its nutrients, particularly amino acids and fatty acids, into body weight. In response to the lactation day, the concentration of all amino acids in milk was higher in colostrum, followed by a dramatic reduction in the present study, which highlighted the importance of colostrum for piglets and more milk protein synthesis at the first day of lactation. The amino acid composition of milk in lactating sows is determined by both ideal dietary amino acid levels and mammary gland metabolism, while the observed variations in this composition can be a consequence of dynamic body tissue mobilization throughout lactation [41]. As litter size increases, so does the requirement of essential amino acids for sows to support milk production and mammary gland tissue; the most limiting of these are typically lysine, threonine, and valine [42]. Among amino acids, lysine requirements have been studied most extensively, given model predictions that they increase substantially for sows nursing large, fast-growing litters [43]. While the literature agrees that a higher dietary lysine intake mitigates sow body weight loss and protein mobilization, its influence on litter growth rate and subsequent reproductive performance is a subject of ongoing debate [44,45]. In the current study, the diet containing 25-OH-D_3_ reduced the content of lysine at the first week of lactation and decreased the plateau levels of lysine, histidine, threonine, aspartic acid, proline, tyrosine, phenylalanine, and isoleucine. A recent study also found that sows fed a diet with 400 g/kg mulberry leaf powder exhibited a higher litter weight gain during lactation despite lower milk concentrations of certain amino acids, including isoleucine, leucine, lysine, and valine [46]. Besides lysine being consistently first-limiting, the requirements for threonine, valine, and leucine are also indispensable for litter performance. Threonine is critical for sows experiencing a low feed intake and high body tissue mobilization, in contrast to valine, which is vital for sows with a high feed intake and minimal mobilization [42]. Serving as a key component of globulins, threonine participates in the production of IgG, which is crucial for neonatal survival and infection prevention [15]. Leucine, which enhances muscle protein synthesis and gut maturation by increasing villus height and crypt depth [16], is also critical in late gestation. It was reported that sows would increase the requirement of leucine and arginine to support the development of fetal and mammary parenchymal tissues during late gestation [41]. Nevertheless, the comparable concentrations of leucine, threonine, and valine in the Ctrl and 25-OH-D_3_ groups do not rule out their contribution to the enhanced litter weight gain. The complex balance and interactive effects within the overall amino acid profile mean that their role cannot be definitively assessed without further study. Given that lactating sows utilize as much as 70% of dietary amino acids for milk protein synthesis [47], there is limited data during the gestation and lactation period for understanding the regulation of dietary interventions in the composition of milk amino acids besides lysine; the precise mechanism by which milk amino acids influence litter performance needs further elucidation.

Milk fatty acids, one of the primary nutrient sources for newborn piglets, play an active role in litter growth. Multiple lines of evidence confirm that vitamin D_3_ plays a critical role in modulating sow milk composition [10,31]. For instance, maternal supplementation with 25-OH-D_3_ has been shown to increase the concentrations of protein and lactose in milk during lactation [10]. Furthermore, dietary manipulation with 25-OH-D_3_ elevates fat content, not only in colostrum [9] but also in mature milk on d 21 of lactation [23]. Some alteration of fatty acids in milk, the mammary glands, and bone marrow was also confirmed in response to dietary 25-OH-D_3_ treatment [22,23]. In this study, the analysis of the milk fatty acid profile showed that saturated fatty acids increased linearly and polyunsaturated fatty acids decreased linearly with advancing lactation, whereas the alterations of total fatty acids and monounsaturated fatty acids were unaffected. Similar patterns have also been noticed in a previous study [23]. Regarding the effects of 25-OH-D_3_ on the fatty acid profile, a previous study found there were no distinct changes in terms of the levels of saturated fatty acids, monounsaturated fatty acids, and polyunsaturated fatty acids in milk [23]. In addition to having no effect on unsaturated fatty acids, dietary 25-OH-D_3_ supplementation increased the concentration of saturated fatty acids on d 21 of lactation [22]. The outcomes of this study revealed that the effect of dietary 25-OH-D_3_ on milk fatty acids occurred in a time-dependent manner. The supplementation of 25-OH-D_3_ initially decreased the content of both saturated and unsaturated fatty acids from d 1 to 7 of lactation. Subsequently, the 25-OH-D_3_-fed group exhibited higher levels of saturated fatty acids on d 14, monounsaturated fatty acids on d 21, and polyunsaturated fatty acids on d 14 and 21 compared to the Ctrl group. This led to an overall increase in fatty acids from d 14 to 21, following the initial decline. The alteration in the fatty acid profile of sow milk had been finely defined to serve as a primary determinant of piglet health, growth, and survival [48]. In addition to fulfilling the high energy demands for thermoregulation and metabolism in newborn piglets, some specific fatty acids, particularly long-chain polyunsaturated fatty acids like arachidonic acid [49], play an indispensable role in the development of piglets. Oleic and linoleic acids are crucial functional components in sow milk, serving as primary energy substrates and playing key roles in piglet health and development [17]. Oleic acid, one of the most abundant fatty acids in mammalian cells, is synthesized de novo from palmitic acid via elongation to stearic acid and subsequent desaturation by stearoyl-CoA desaturase 1 (SCD1), the key enzyme catalyzing the formation of the double bond [50]. Linoleic acid serves as a precursor for the biosynthesis of longer-chain fatty acids, including 11-eicosadienoic acid, via fatty acid elongase-5 [50]. Dietary oleic and linoleic acid supplementations were related to the growth and survivability of piglets, serving as the preferred substrates for energy metabolism in pigs [17]. Beyond their energetic value, oleic and linoleic acids offer distinct health benefits for piglets. Oleic acid demonstrates anti-inflammatory properties and confers protection against metabolic diseases such as insulin resistance, which are linked to enhanced fatty acid oxidation in skeletal muscle [18]. Linoleic acid was observed to attenuate mucosal damage induced by colitis in pigs [19]. Importantly, the presence of linoleic acid in milk is closely associated with improved survivability and body weight gain in suckling piglets [51]. The results in this study also showed that sows fed diets with 25-OH-D_3_ had increased ratios of unsaturated: saturated fatty acids with the extent of lactation. This was characterized by decreases in saturated fatty acids, such as myristic, palmitic, and stearic acid. The initial phase also saw a reduction in unsaturated fatty acids, including oleic, linoleic, 11-eicosenoic, and arachidonic acid; however, a subsequent shift occurred where their contents rose and remained elevated until d 21. The alteration of the milk fatty acid profile of sows through 25-OH-D_3_ supplementation has direct implications for piglet outcomes. It was established that the precise temporal dynamics of fatty acids in piglets constitute a sophisticated evolutionary adaptation, ensuring that the right fuel and structural components are available to overcome sequential physiological hurdles like the neonatal energy–thermal crisis, the development of the brain and immune system, and the challenges of weaning. In this study, the initial shift in the fatty acid ratio and the subsequent increase in beneficial long-chain unsaturated fatty acids, such as oleic, linoleic, and arachidonic acid, likely provide piglets with a superior energy source and essential building blocks for organ development.

There are two main limitations in this study. The first is the methodology. Genes, proteins related to both amino acids and glycolipid metabolism, and related rate-limiting enzymes should be determined to further define the alteration in the milk fatty acid or amino acid profile due to dietary 25-OH-D_3_ supplementation, which was not performed in our study. Furthermore, the study lacked data on how 25-OH-D_3_ manipulation affects key components of the IGF-1 axis, including IGF-1 and GH. This omission prevents a mechanistic understanding of how dietary 25-OH-D_3_ supplementation alters milk composition and sow performance. Secondly, statistical power was constrained by unequal variances and an insufficient sample size. To address these limitations in future work, mid-infrared spectroscopy could be employed to predict milk components, especially amino acids and fatty acids profile [52]. In addition, a partial least squares regression might directly model the relationship between milk composition (predictor) and sow performance (response), identifying the most influential variables [53]. Therefore, we admit the possibility that some of our conclusions may include an overestimation or underestimation of the roles of dietary 25-OH-D_3_ or milk composition in sow performance. Additional research will be essential to address this limitation.

## 5. Conclusions

The findings of the current study indicated that a dietary supplementation with 50 μg/kg 25-OH-D_3_ improved weaning weight and weight gain and kept sows in better physical condition. In response to the lactation day, amino acid levels were highest in colostrum and then fell. This drop was faster in sows given 25-OH-D_3_, resulting in lower amino acid levels by the end of lactation. Over the lactation period, saturated fats increased while polyunsaturated fats decreased. The 25-OH-D_3_ supplement first lowered all fat levels, then later (l4 to 21 d of lactation) increased specific beneficial fats, especially oleic and linoleic acids and arachidonic acid. These findings support the hypothesis that maternal supplementation of 25-OH-D_3_ in gestation and lactation diets could alter milk composition, which appeared to be connected to the improved piglet growth.

## Figures and Tables

**Figure 1 animals-15-03160-f001:**
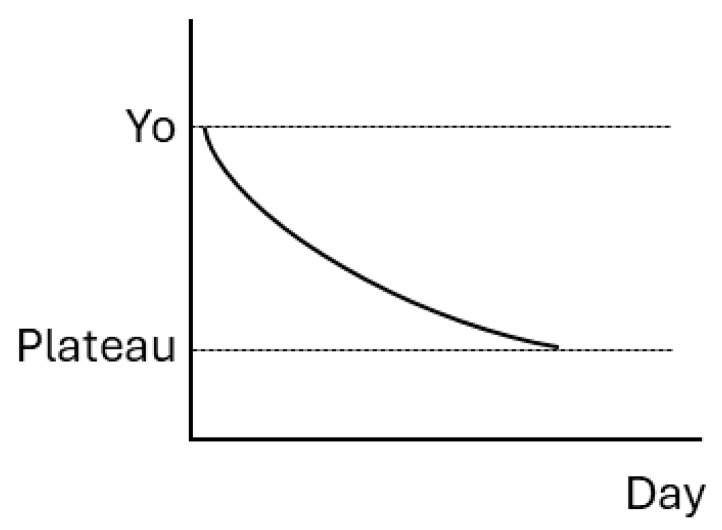
The model of one-phase decay.

**Figure 2 animals-15-03160-f002:**
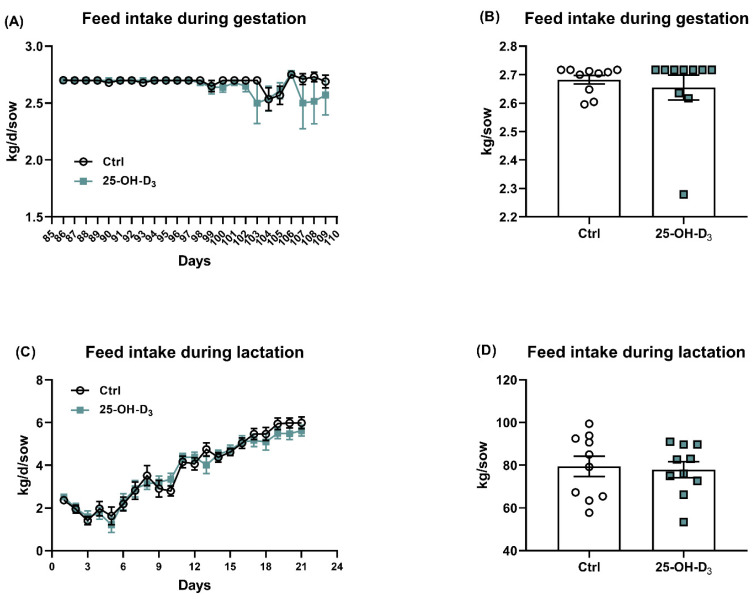
Effects of dietary 25-OH-D_3_ supplementation on the feed intake during gestation and lactation. Data was analyzed using an independent sample *t*-test procedure with Bonferroni correction. The circles and squares represent the corresponding samples in each treatment. Statistical significance was set at *p* < 0.05. Results are expressed as means and their standard errors.

**Figure 3 animals-15-03160-f003:**
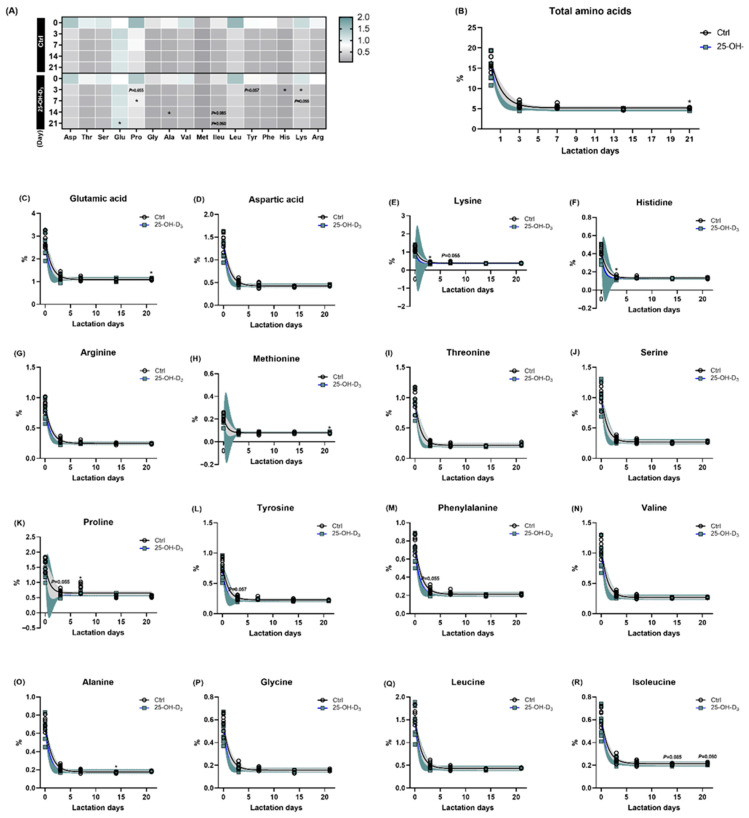
The amino acids of milk in response to lactation days and dietary supplementation of 25-OH-D_3_. The circles and squares represent the corresponding samples in each treatment. Supplementation (**A**) and lactation days (**B**), as well as both dietary 25-OH-D_3_ manipulation and lactation days on each aminol acid (**C**–**R**). The circles and squares represent the corresponding samples in each treatment. * Indicates a significant difference between the Ctrl and 25-OH-D_3_ treatments at *p* < 0.05. Results are expressed as means and their standard errors.

**Figure 4 animals-15-03160-f004:**
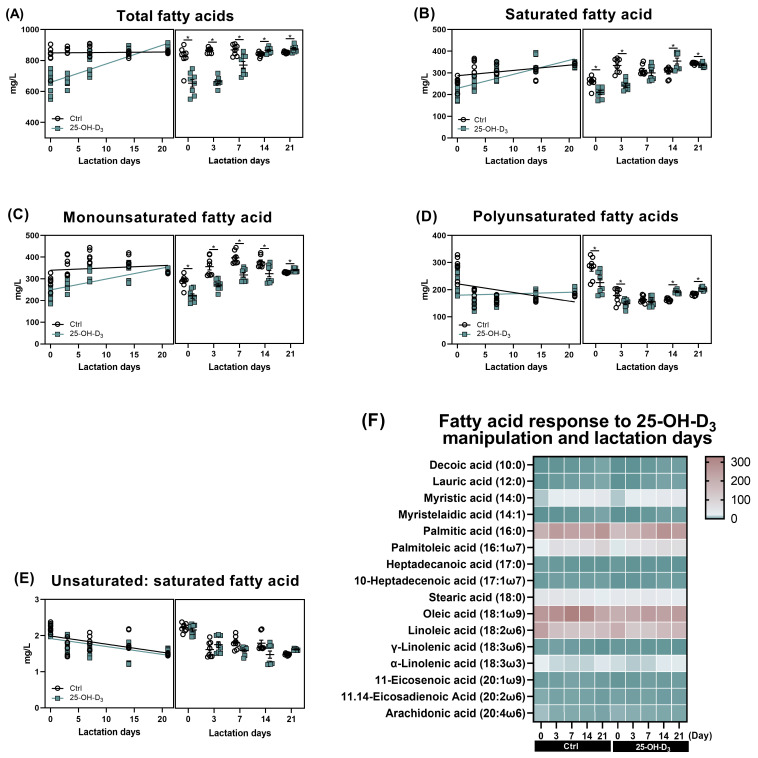
Effects of lactation days and dietary supplementation of 25-OH-D_3_ on fatty acids of milk. The alterations in total fatty acids (**A**), saturated fatty acid (**B**), monounsaturated fatty acid (**C**), polyunsaturated fatty acid (**D**), and their ratio (**E**), as well as each fatty acid (**F**) response to both dietary 25-OH-D_3_ manipulation and lactation days. The circles and squares represent the corresponding samples in each treatment. * Indicates a significant difference between the Ctrl and 25-OH-D_3_ treatments at *p* < 0.05. Results are expressed as means and their standard errors.

**Figure 5 animals-15-03160-f005:**
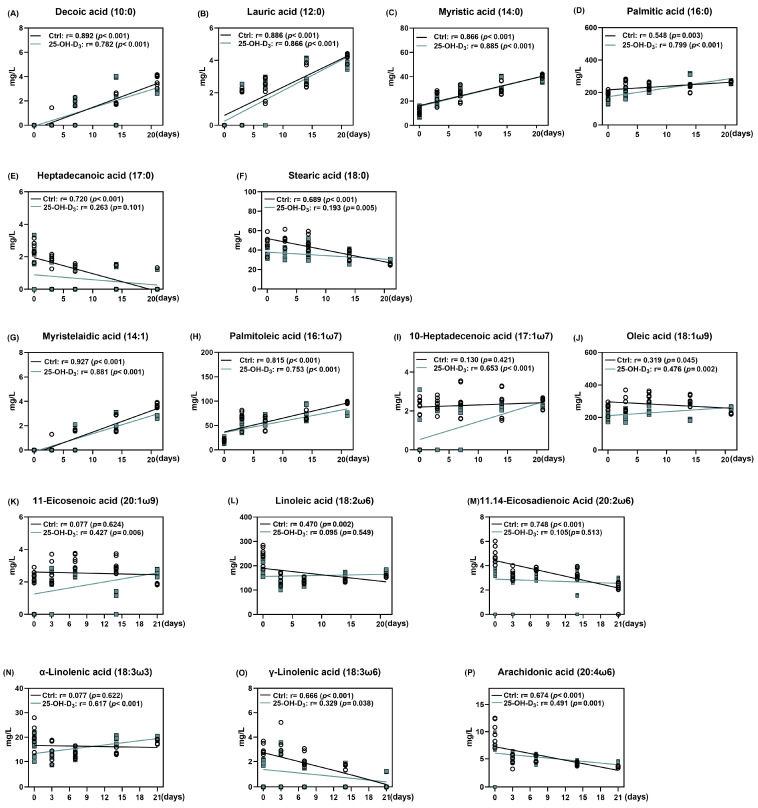
The alterations of each fatty acid in milk in response to lactation days. The circles and squares represent the corresponding samples in each treatment. Results are expressed as means and their standard errors.

**Figure 6 animals-15-03160-f006:**
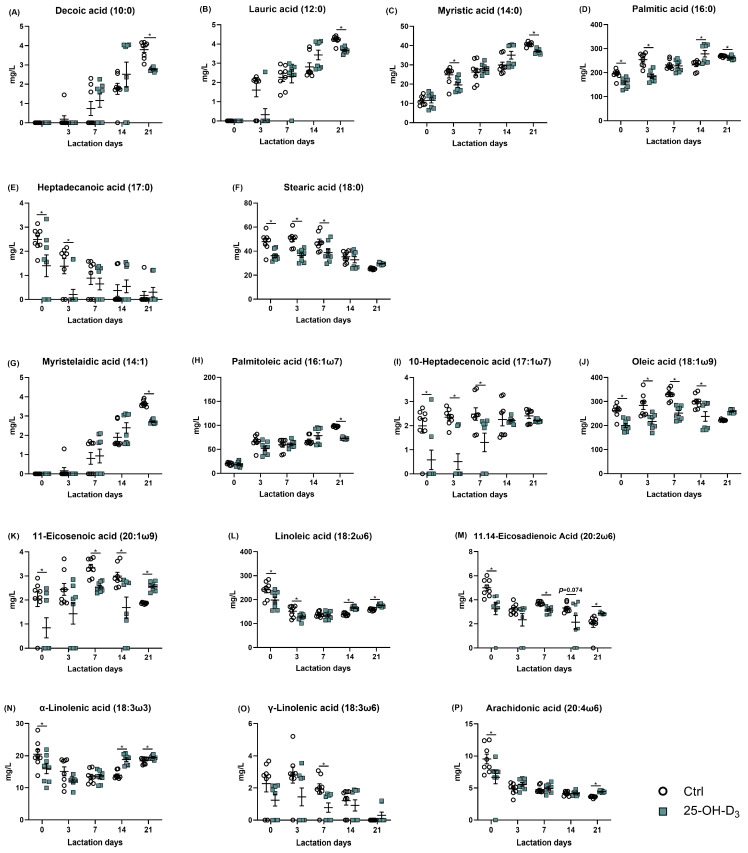
The alterations of each fatty acid in milk in response to dietary supplementation of 25-OH-D_3_. The circles and squares represent the corresponding samples in each treatment. * Indicates a significant difference between the Ctrl and 25-OH-D_3_ treatments at *p* < 0.05. Results are expressed as means and their standard errors.

**Table 1 animals-15-03160-t001:** Nutrient composition of the gestation and lactation diets (as-fed basis).

Items	Lactation Diet		Gestation Diet	
	Ctrl	25-OH-D_3_	Ctrl	25-OH-D_3_
Ingredient				
Corn	58.600	58.600	45.319	45.319
Soya bean meal, 43% crude protein	11.30	11.30	12.59	12.59
Fermented soybean meal			5	5
Extruded soybean			5	5
Soybean skin	14	14	3	3
Wheat bran	10.5	10.5	1.3	1.3
Barley with husk			15	15
wheat flour			6	6
Limestone	3.1	3.1	1.3	1.3
Dicalcium phosphate	0.70	0.70	1.31	1.31
Sodium chloride	0.35	0.35	0.4	0.4
Sodium bicarbonate	0.2	0.2	0.2	0.2
Soybean oil			0.5	0.5
L-Lysine HCL	0.256	0.256	0.670	0.670
L-Threonine	0.082	0.082	0.23	0.23
DL-Methionine	0.066	0.066	0.100	0.100
L-Tryptophan	0.039	0.039	0.04	0.04
Valine	0.112	0.112	0.175	0.175
Choline chloride	0.10	0.10	0.10	0.10
25-OH-D_3_		0.02		0.02
Zeolite powder	0.02		0.02	
Vitamin premix ^1^	0.05	0.05	0.04	0.04
Mineral premix ^2^	0.38	0.38	0.38	0.38
Other premix ^3^	0.145	0.145	1.326	1.326
Total	100	100	100	100
Nutrient composition, %				
Metabolic energy, kcal/kg	2994.0	2994.0	3328.0	3328.0
Crude protein	12.5	12.5	17.3	17.3
Ether extract	2.71	2.71	3.96	3.96
Crude fiber	8.00	8.00	4.04	4.04
Calcium	1.44	1.44	0.82	0.82
Total phosphorus	0.5	0.5	0.63	0.63
Standard ileal digestibility lysine	0.6	0.6	1.06	1.06
Standard ileal digestibility methionine	0.23	0.23	0.34	0.34
Standard ileal digestibility tryptophan	0.11	0.11	0.20	0.20

^1^ Supplied per kg of diet: Zn, 94.2 mg as ZnSO_4_; Fe, 131.5 mg as FeSO_4_·H_2_O; Mn 50 mg, as MnSO_4_; Cu, 7 mg as CuSO_4_·5H_2_O; I, 0.30 mg as CaI_2_O_6_; Se, 0.30 mg as NaSeO_3_; Mg, 1.8 g as MgO; and Cr as CrCl_3_. ^2^ Supplied per kg of diet: vitamin A, 6720 IU; vitamin D_3_, 3200 IU; vitamin E, 80 IU; vitamin K (as menadione sodium bisulfite complex), 3.21 mg; thiamine, 1.78 mg; riboflavin, 8 mg; pantothenic acid (as d-calcium pantothenate), 20.38 mg; niacin, 30.12 mg; vitamin B12, 20 µg; d-biotin, 0.50 mg; folic acid, 5.83 mg; and pyridoxine, 4.47 mg. ^3^ Provided the following quantities in gestation diet: wine yeast extract, 0.5%; linseed oil, 0.5%; chaff, 0.1%; acidifier, 0.05%; mold inhibitor, 0.05%; mycotoxin adsorbent, 0.05%; 4,7-dihydroxyisoflavone, 0.02%; phytase, 0.015%; plant essential oil, 0.014%; plant extract, 0.012%; chromium pyrochlore, 0.01%; and selenium yeast, 0.005%. The following quantities were provided in lactation diet: antimicrobial agent, 0.08%, microbial preparation, 0.03%, antioxidant, 0.02%; and phytase, 0.015%.

**Table 2 animals-15-03160-t002:** Sow body condition and tear stain scoring system.

Item		Definition
Sow body condition score		
1	Emaciated	Back bone and ribs visible
2	Thin	Back bone and ribs can be easily felt
3	Fit	Back bone and ribs can barely be felt
4	Fat	Back bone and ribs cannot be felt
5	Very fat	Obviously overweight
Tear stain score		
0		No signs of staining
1		Barely detectable staining
2		Stained area <50% area compared with size of an eye area
3		Staining of 50–100% of total eye area
4		Stained area >100% of total eye area
5		Staining >100% of total eye area and staining extend below the mouth line

**Table 3 animals-15-03160-t003:** The tear stain and body condition of sow in response to dietary supplementation of 25-OH-D_3_.

Item	Treatment		*p*-Value
	Ctrl	25-OH-D_3_	
Tear stain score			
93 d of gestation	2.4 ± 0.84	2.5 ± 0.85	0.795
101 d of gestation	2.25 ± 0.42	1.65 ± 0.67	0.030
108 d of gestation	2.3 ± 0.54	2 ± 0.67	0.283
Body condition score			
93 d of gestation	3.05 ± 0.28	2.95 ± 0.16	0.347
101 d of gestation	3.2 ± 0.35	3.15 ± 0.34	0.749
108 d of gestation	3.15 ± 0.24	3.15 ± 0.34	0.994
Backfat thickness (Digital)			
85 d of gestation	17.6 ± 2.27	18.5 ± 1.41	0.32
Delivery	18.9 ± 1.6	17.75 ± 1.98	0.205
20 d of lactation	14.7 ± 1.83	14.88 ± 2.59	0.874
Loss of backfat during lactation	4.2 ± 1.62	3.4 ± 2.07	0.349
Backfat thickness (Digital)			
85 d of gestation	19.85 ± 1.78	19.81 ± 1.94	0.967
Delivery	21.65 ± 1.78	22.63 ± 1.81	0.27
20 d of lactation	15.81 ± 1.54	16.69 ± 2.66	0.425
Loss of backfat during lactation	5.84 ± 1.73	6.1 ± 1.41	0.717

Data were analyzed using an independent sample *t*-test procedure with Bonferroni correction after the normal distribution test. Statistical significance was set at *p* < 0.05. Results are expressed as means and their standard errors.

**Table 4 animals-15-03160-t004:** Effects of dietary 25-OH-D_3_ supplementation on performance of sows during gestation.

Item	Ctrl	25-OH-D_3_	*p*-Value
Gestation, day	116.4 ± 0.7	116.2 ± 1.14	0.642
Body weight of sows, kg			
Gestation			
0 d	140.7 ± 3.51	140.8 ± 4.02	0.768
110 d	197.6 ± 2.83	199.1 ± 2.12	0.914
Change	+56.8 ± 1.43	+58.4 ± 0.96	0.712
Lactation			
0 d	182.9 ± 3.65	182.3 ± 4.94	0.958
21 d	179.1 ± 4.77	177.6 ± 3.08	0.665
Change	−3.8 ± 1.68	−4.6 ± 1.09	0.269
Change litter size, *n*			
Live-born	12.9 ± 3.54	11.0 ± 2.49	0.184
Stillborn	0.5 ± 0.71	0.6 ± 0.84	0.777
Mummy	0.0 ± 0.00	0.2 ± 0.63	0.343
Deformity	0.1 ± 0.32	0.2 ± 0.63	0.662
Total born	13.4 ± 3.69	12 ± 2.67	0.345
Litter weight at birth (not including deformity), kg	18.31 ± 3.78	16.04 ± 4.42	0.234
Body weight of piglets at birth, kg	1.47 ± 0.24	1.46 ± 0.24	0.916

Data were analyzed using an independent sample *t*-test procedure with Bonferroni correction after the normal distribution test. Statistical significance was set at *p* < 0.05. Results are expressed as means and their standard errors.

**Table 5 animals-15-03160-t005:** Effects of dietary 25-OH-D_3_ supplementation on performance of sows during lactation.

Item	Ctrl	25-OH-D_3_	*p*-Value
Litter size, n			
Adjusted live-born	12.9 ± 0.88	11.0 ± 0.67 *	<0.001
Weaned	12.5 ± 1.27	10.3 ± 1.25 *	0.064
Survivability, %	96.79 ± 5.71	93.45 ± 7.57	0.455
Initial body weight	1.47 ± 0.24	1.46 ± 0.24	0.872
Weaning body weight, kg	5.50 ± 0.77	6.16 ± 0.48 *	0.031
Body gain, kg/piglet	4.02 ± 0.64	4.68 ± 0.40 *	0.016

Data were analyzed using an independent sample *t*-test procedure with Bonferroni correction after the normal distribution test. * Indicates a significant difference between the Ctrl and 25-OH-D_3_ treatments at *p* < 0.05. Results are expressed as means and their standard errors.

**Table 6 animals-15-03160-t006:** Effects of dietary 25-OH-D_3_ supplementation on amino acid profile of milk (g/100 g milk).

Item	Lactation Days					Best-Fit Values		
	0	3	7	14	21	Plateau	K	r
Glutamic acid								
Ctrl	2.85 ± 0.34	1.23 ± 0.13	1.13 ± 0.08	1.05 ± 0.04	1.09 ± 0.03	1.086	0.825	0.975
25-OH-D_3_	2.65 ± 0.47	1.17 ± 0.11	1.11 ± 0.03	1.08 ± 0.06	1.15 ± 0.01 *	1.113	1.120	0.948
*p*-value	0.340	0.295	0.544	0.251	<0.001			
Aspartic acid								
Ctrl	1.42 ± 0.16	0.51 ± 0.05	0.43 ± 0.05	0.41 ± 0.01	0.43 ± 0.01	0.396	1.050	0.980
25-OH-D_3_	1.31 ± 0.25	0.48 ± 0.05	0.44 ± 0.01	0.42 ± 0.01	0.44 ± 0.01	0.388	1.940	0.942
*p*-value	0.283	0.222	0.681	0.554	0.377			
Lysine								
Ctrl	1.20 ± 0.14	0.43 ± 0.03	0.43 ± 0.03	0.38 ± 0.01	0.39 ± 0.01	0.397	1.048	0.980
25-OH-D_3_	1.09 ± 0.23	0.39 ± 0.03 *	0.40 ± 0.01	0.38 ± 0.01	0.38 ± 0.01	0.388	1.937	0.942
*p*-value	0.283	0.030	0.055	0.559	0.483			
Histidine								
Ctrl	0.44 ± 0.05	0.15 ± 0.01	0.14 ± 0.01	0.13 ± 0.00	0.13 ± 0.01	0.134	1.018	0.984
25-OH-D_3_	0.40 ± 0.08	0.13 ± 0.01 *	0.14 ± 0.01	0.13 ± 0.01	0.13 ± 0.00	0.131	1.687	0.949
*p*-value	0.199	0.033	0.212	0.810	0.411			
Arginine								
Ctrl	0.89 ± 0.10	0.30 ± 0.04	0.26 ± 0.02	0.24 ± 0.01	0.24 ± 0.01	0.247	0.825	0.984
25-OH-D_3_	0.79 ± 0.15	0.28 ± 0.03	0.26 ± 0.01	0.24 ± 0.00	0.25 ± 0.01	0.250	0.983	0.956
*p*-value	0.153	0.216	0.785	0.647	0.109			
Methionine								
Ctrl	0.22 ± 0.03	0.08 ± 0.01	0.08 ± 0.01	0.08 ± 0.01	0.08 ± 0.00	0.079	1.344	0.965
25-OH-D_3_	0.20 ± 0.04	0.08 ± 0.01	0.08 ± 0.01	0.08 ± 0.00	0.08 ± 0.01	0.081	1.885	0.925
*p*-value	0.329	1.00	0.560	0.591	0.490			
Threonine								
Ctrl	0.98 ± 0.16	0.26 ± 0.03	0.22 ± 0.03	0.21 ± 0.01	0.23 ± 0.02	0.216	0.964	0.975
25-OH-D_3_	0.90 ± 0.21	0.24 ± 0.02	0.23 ± 0.01	0.20 ± 0.01	0.22 ± 0.01	0.215	1.048	0.951
*p*-value	0.385	0.283	0.389	0.248	0.245			
Serine								
Ctrl	1.06 ± 0.16	0.32 ± 0.04	0.28 ± 0.03	0.26 ± 0.01	0.28 ± 0.01	0.270	0.913	0.975
25-OH-D_3_	1.00 ± 0.23	0.30 ± 0.03	0.29 ± 0.01	0.26 ± 0.01	0.28 ± 0.01	0.275	1.066	0.948
*p*-value	0.496	0.334	0.376	0.400	0.591			
Proline								
Ctrl	1.56 ± 0.22	0.67 ± 0.08	0.85 ± 0.14	0.56 ± 0.04	0.54 ± 0.03	0.648	1.144	0.916
25-OH-D_3_	1.37 ± 0.26	0.60 ± 0.06	0.66 ± 0.03 *	0.57 ± 0.04	0.55 ± 0.01	0.595	1.636	0.933
*p*-value	0.138	0.055	0.007	0.766	0.199			
Tyrosine								
Ctrl	0.82 ± 0.11	0.26 ± 0.03	0.25 ± 0.02	0.22 ± 0.02	0.22 ± 0.01	0.228	0.924	0.978
25-OH-D_3_	0.73 ± 0.17	0.24 ± 0.02	0.24 ± 0.01	0.21 ± 0.01	0.22 ± 0.01	0.222	1.215	0.943
*p*-value	0.255	0.057	0.149	0.452	0.269			
Phenylalanine								
Ctrl	0.77 ± 0.09	0.26 ± 0.03	0.23 ± 0.02	0.20 ± 0.01	0.21 ± 0.01	0.214	0.801	0.982
25-OH-D_3_	0.70 ± 0.14	0.24 ± 0.02	0.23 ± 0.01	0.20 ± 0.01	0.21 ± 0.00	0.213	1.018	0.953
*p*-value	0.138	0.055	0.707	0.766	0.199			
Valine								
Ctrl	1.12 ± 0.15	0.33 ± 0.04	0.28 ± 0.03	0.26 ± 0.01	0.27 ± 0.01	0.268	0.888	0.981
25-OH-D_3_	0.99 ± 0.23	0.30 ± 0.03	0.28 ± 0.01	0.26 ± 0.01	0.27 ± 0.00	0.270	1.030	0.945
*p*-value	0.229	0.165	0.904	0.727	0.441			
Alanine								
Ctrl	0.72 ± 0.07	0.22 ± 0.03	0.19 ± 0.02	0.17 ± 0.01	0.18 ± 0.00	0.178	0.816	0.987
25-OH-D_3_	0.66 ± 0.14	0.2 ± 0.02	0.19 ± 0	0.18 ± 0.01 *	0.18 ± 0.00	0.181	1.014	0.955
*p*-value	0.285	0.111	0.590	0.035	1.000			
Glycine								
Ctrl	0.57 ± 0.07	0.19 ± 0.02	0.17 ± 0.01	0.15 ± 0.01	0.16 ± 0.01	0.158	0.834	0.979
25-OH-D_3_	0.52 ± 0.11	0.18 ± 0.02	0.17 ± 0.01	0.15 ± 0.01	0.16 ± 0.00	0.158	1.016	0.95
*p*-value	0.262	0.166	0.829	0.435	0.104			
Leucine								
Ctrl	1.59 ± 0.18	0.52 ± 0.06	0.44 ± 0.04	0.42 ± 0.01	0.44 ± 0.01	0.431	0.867	0.985
25-OH-D_3_	1.44 ± 0.31	0.48 ± 0.04	0.45 ± 0.02	0.42 ± 0.01	0.44 ± 0.01	0.432	1.009	0.948
*p*-value	0.258	0.190	0.807	0.805	0.693			
Isoleucine								
Ctrl	0.63 ± 0.07	0.26 ± 0.03	0.22 ± 0.02	0.21 ± 0.01	0.22 ± 0.01	0.214	0.768	0.979
25-OH-D_3_	0.58 ± 0.12	0.24 ± 0.02	0.22 ± 0.01	0.20 ± 0.01	0.21 ± 0.00	0.211	0.864	0.942
*p*-value	0.292	0.221	0.751	0.085	0.060			
Total amino acids								
Ctrl	16.84 ± 2.05	5.99 ± 0.65	5.59 ± 0.43	4.94 ± 0.18	5.09 ± 0.12	5.190	0.881	0.979
25-OH-D_3_	15.30 ± 3.12	5.54 ± 0.52	5.37 ± 0.11	4.98 ± 0.11	4.62 ± 0.05 *	4.977	0.957	0.950
*p*-value	0.266	0.148	0.212	0.667	0.000			

* Indicates a significant difference between the Ctrl and 25-OH-D_3_ treatments at *p* < 0.05. Results are expressed as means and their standard errors.

**Table 7 animals-15-03160-t007:** Effects of dietary 25-OH-D_3_ supplementation on fatty acid profile of milk (g/kg milk fat).

Item	Lactation Days					Best-Fit Values	
	0	3	7	14	21	r	*p*-Value
Saturated fatty acid (SFA)							
Decanoic acid (10:0)							
Ctrl	0.00 ± 0.00	0.00 ± 0.00	0.74 ± 1.03	1.76 ± 0.81	3.79 ± 0.43	0.892	<0.001
25-OH-D_3_	0.00 ± 0.00	0.00 ± 0.00	1.15 ± 0.97	2.52 ± 1.78	2.76 ± 0.10 *	0.782	<0.001
*p*-value	-	-	0.423	0.300	<0.001		
Lauric acid (12:0)							
Ctrl	0.00 ± 0.00	1.61 ± 0.99	2.26 ± 0.60	2.81 ± 0.61	4.22 ± 0.20	0.886	<0.001
25-OH-D_3_	0.00 ± 0.00	0.32 ± 0.90 *	2.32 ± 0.96	3.43 ± 0.71	3.67 ± 0.14 *	0.866	<0.001
*p*-value	-	0.017	0.891	0.084	<0.001		
Myristic acid (14:0)							
Ctrl	11.39 ± 2.30	24.81 ± 4.54	26.29 ± 5.66	29.88 ± 4.44	40.73 ± 1.02	0.866	<0.001
25-OH-D_3_	11.57 ± 3.75	19.56 ± 3.50 *	27.98 ± 2.52	35.01 ± 5.58	36.97 ± 1.22 *	0.885	<0.001
*p*-value	0.910	0.022	0.460	0.062	<0.001		
Palmitic acid (16:0)							
Ctrl	194.23 ± 18.41	254.13 ± 27.66	232.83 ± 18.62	232.31 ± 21.27	268.96 ± 3.21	0.548	0.003
25-OH-D_3_	157.18 ± 20.75 *	184.99 ± 22.04 *	227.86 ± 24.28	278.55 ± 38.17 *	261.79 ± 6.84 *	0.799	<0.001
*p*-value	0.002	<0.001	0.653	0.012	0.023		
Heptadecanoic acid (17:0)							
Ctrl	2.49 ± 0.48	1.38 ± 0.89	0.89 ± 0.75	0.37 ± 0.69	0.17 ± 0.47	−0.720	<0.001
25-OH-D_3_	1.40 ± 1.28 *	0.21 ± 0.59 *	0.64 ± 0.69	0.55 ± 0.75	0.30 ± 0.56	−0.263	0.101
*p*-value	0.050	0.009	0.509	0.640	0.608		
Stearic acid (18:0)							
Ctrl	47.88 ± 7.57	50.09 ± 6.27	47.46 ± 7.30	35.32 ± 4.52	25.32 ± 0.64	−0.689	<0.001
25-OH-D_3_	36.18 ± 4.37 *	36.46 ± 5.04 *	38.77 ± 8.03 *	32.82 ± 7.15	29.47 ± 0.83 *	−0.193	0.005
*p*-value	0.003	0.003	0.040	0.419	<0.001		
Monounsaturated fatty acid (MUFA)							
Myristelaidic acid (14:1)							
Ctrl	0.00 ± 0.00	0.16 ± 0.46	0.80 ± 0.86	1.90 ± 0.62	3.65 ± 0.16	0.927	<0.001
25-OH-D_3_	0.00 ± 0.00	0.00 ± 0.00	0.93 ± 1.01	2.39 ± 0.76	2.70 ± 0.10 *	0.881	<0.001
*p*-value	-	0.351	0.788	0.181	<0.001		
Palmitoleic acid (16:1ω7)							
Ctrl	20.19 ± 2.48	66.28 ± 13.36	59.25 ± 13.10	68.39 ± 8.54	98.06 ± 1.51	0.815	<0.001
25-OH-D_3_	18.28 ± 5.02	52.49 ± 12.71	60.22 ± 7.12	78.73 ± 16.96	72.90 ± 2.02 *	0.753	<0.001
*p*-value	0.356	0.053	0.858	0.154	<0.001		
10-Heptadecenoic acid (17:1ω7)							
Ctrl	2.00 ± 0.87	2.35 ± 0.34	2.49 ± 0.73	2.26 ± 0.75	2.41 ± 0.29	0.130	0.421
25-OH-D_3_	0.58 ± 1.15 *	0.51 ± 0.94 *	1.30 ± 1.08 *	2.21 ± 0.12	2.21 ± 0.06	0.653	<0.001
*p*-value	0.016	0.001	0.025	0.863	0.102		
Oleic acid (18:1ω9)							
Ctrl	262.31 ± 26.84	283.84 ± 47.07	329.14 ± 23.70	300.12 ± 28.37	222.76 ± 2.62	−0.319	0.045
25-OH-D_3_	200.35 ± 23.40 *	216.95 ± 30.35 *	252.3 ± 28.02 *	238.09 ± 56.02 *	258.82 ± 7.41 *	0.476	0.002
*p*-value	<0.001	0.006	<0.001	0.018	<0.001		
11-Eicosenoic acid (20:1ω9)							
Ctrl	2.04 ± 0.88	2.44 ± 0.69	3.35 ± 0.37	2.99 ± 0.46	1.87 ± 0.03	−0.077	0.624
25-OH-D_3_	0.85 ± 1.17 *	1.43 ± 1.23	2.53 ± 0.2 *	1.69 ± 1.22 *	2.56 ± 0.18 *	0.427	0.006
*p*-value	0.038	0.067	<0.001	0.020	<0.001		
Polyunsaturated fatty acids (PUFA)							
Linoleic acid (18:2ω6)							
Ctrl	241.27 ± 34.72	151.76 ± 23.29	138.55 ± 10.67	138.86 ± 5.43	158.31 ± 4.09	−0.470	0.002
25-OH-D_3_	198.40 ± 35.62 *	127.52 ± 13.13 *	133.30 ± 15.48	165.03 ± 5.92 *	175.84 ± 5.23 *	−0.095	0.549
*p*-value	0.029	0.026	0.444	<0.001	<0.001		
11.14-Eicosadienoic acid (20:2ω6)							
Ctrl	5.00 ± 0.70	3.23 ± 0.42	3.74 ± 0.12	3.33 ± 0.38	2.01 ± 0.84	−0.748	<0.001
25-OH-D_3_	3.26 ± 1.38 *	2.34 ± 1.46	3.16 ± 0.28	2.13 ± 1.59	2.86 ± 0.09 *	−0.105	0.513
*p*-value	0.009	0.138	<0.001	0.074	0.024		
α-Linolenic acid (18:3ω3)							
Ctrl	20.49 ± 4.16	15.13 ± 4.04	13.46 ± 2.08	13.94 ± 1.22	18.40 ± 0.93	−0.077	0.622
25-OH-D_3_	15.93 ± 4.25 *	12.14 ± 1.62	13.45 ± 1.92	18.83 ± 1.59 *	19.47 ± 0.59 *	0.617	<0.001
*p*-value	0.048	0.083	0.996	<0.001	0.018		
γ-Linolenic acid (18:3ω6)							
Ctrl	2.27 ± 1.45	2.81 ± 1.42	1.94 ± 0.93	1.21 ± 0.77	0.00 ± 0.00	−0.666	<0.001
25-OH-D_3_	1.24 ± 1.04	1.44 ± 1.57	0.78 ± 0.83 *	0.92 ± 0.98	0.30 ± 0.55 *	−0.329	0.038
*p*-value	0.128	0.089	0.020	0.522	0.171		
Arachidonic acid (20:4ω6)							
Ctrl	9.53 ± 2.14	4.85 ± 0.87	4.77 ± 0.56	4.09 ± 0.32	3.62 ± 0.14	−0.674	<0.001
25-OH-D_3_	6.68 ± 2.88 *	5.58 ± 0.83	4.93 ± 0.70	4.20 ± 0.33	4.43 ± 0.13 *	−0.491	0.001
*p*-value	0.043	0.110	0.601	0.538	<0.001		
Fatty acid profile							
Total fatty acids							
Ctrl	826.61 ± 70.06	865.92 ± 16.59	869.97 ± 34.58	841.1 ± 14.32	854.27 ± 5.75	0.054	0.726
25-OH-D_3_	652.58 ± 71.57 *	661.93 ± 31.78 *	771.82 ± 66.09 *	867.5 ± 19.02 *	877.33 ± 22.17 *	0.860	<0.001
*p*-value	<0.001	<0.001	0.004	0.008	0.022		
SFA							
Ctrl	256.88 ± 24.56	332.51 ± 32.15	310.95 ± 22.56	302.45 ± 24.71	343.19 ± 4.22	0.496	0.001
25-OH-D_3_	207.02 ± 25.30 *	241.54 ± 22.13 *	298.72 ± 34.33	353.16 ± 39.35 *	334.95 ± 8.85	0.805	<0.001
*p*-value	0.001	<0.001	0.416	0.010	0.093		
MUFA							
Ctrl	288.52 ± 26.54	355.62 ± 43.23	396.00 ± 28.77	377.03 ± 25.25	328.74 ± 2.81	0.176	0.280
25-OH-D_3_	220.05 ± 28.46 *	271.38 ± 24.31 *	317.48 ± 29.77 *	323.29 ± 41.03 *	339.48 ± 8.99 *	0.745	<0.001
*p*-value	<0.001	0.001	<0.001	0.009	0.011		
PUFA							
Ctrl	281.21 ± 38.89	177.78 ± 27.54	163.01 ± 12.36	161.62 ± 5.67	182.34 ± 4.56	−0.494	0.001
25-OH-D_3_	225.51 ± 40.68 *	149.02 ± 15.26 *	155.62 ± 17.71	191.05 ± 6.72 *	202.90 ± 5.87 *	0.013	0.475
*p*-value	0.014	0.026	0.351	<0.001	<0.001		
Unsaturated to saturated fatty acid							
Ctrl	2.22 ± 0.11	1.62 ± 0.21	1.81 ± 0.17	1.80 ± 0.23	1.49 ± 0.03	−0.581	<0.001
25-OH-D_3_	2.16 ± 0.14	1.76 ± 0.23	1.59 ± 0.11 *	1.48 ± 0.28 *	1.62 ± 0.02 *	−0.570	<0.001
*p*-value	0.339	0.250	0.011	0.029	<0.001		

* indicates a significant difference between the Ctrl and 25-OH-D_3_ treatments at *p* < 0.05. Results are expressed as means and their standard errors.

## Data Availability

The original contributions presented in this study are included in the article. Further inquiries can be directed to the corresponding author(s).

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
