# Peer review of "Maternal 25-Hydroxycholecalciferol Supplementation Dynamically Altered Milk Fatty Acid and Amino Acid Profiles and Improves Sow Performance"

_animals, 2025, doi:10.3390/ani15213160_

Round 1

Reviewer 1 Report

Comments and Suggestions for Authors

This study evaluate the effects of maternal 25-hydroxycholecalciferol (25-OH-D3) supplementation on sow, from day 85 of gestation until they are weaned. They analyze the performance of sows, the growth of offspring, and the amino acid and fatty acid profile of the milk.

2. Materials and Methods

2.1. Animals, diets and management

  • Line 112: "at 85 d of pregnancy were allocated into two groups (10 sows/group)" The expression isn't correct. The sows aren't pregnant, they're gestating.
  • Line 115: "Dietary treatments were administered throughout gestation and lactation, with weaning performed on d 21 of lactation." The treatment is not during the entire gestation, it starts from day 85.

2.2. Back-fat thickness and sow body condition score

  • Line 139: "Two points located 65 mm from the dorsal midline, aligned with the last rib, were shaved on standing sows". Bad translation. Back-fat is not measured at two points, but at a single point called P2.

2.4. Performance of sows

  • I'm missing information on how food is administered to sows, both during gestation and lactation. How often do they eat? What are the feeders like? How is their intake measured? Therefore, how is each sow's intake calculated and recorded?
  • Line 165: "Litter birth weight was measured, and individual piglet birth weights were calculated by dividing total litter weight by the number of live-born piglets". I understand that the litter weight, and therefore the individual weight of the piglets, is the weight of each sow once she has farrowed.
    After this, are piglets moved between sows? If so, birth weight is one thing, and post-movement weight is another.
    This latter weight should be taken into account when calculating growth and weight gain. 

3. Results

3.1. Tear stain score and body condition

  • Figure 1. The dates described in the methodology and those represented in the results do not match. For example, in methods back-fat thickness was measured on d 93, 101, and 108 of gestation, as well as d 20 of lactation. In figure there are d 85, delivery and d 20 of lactation. nothing to do with that.

3.2. Sow performance

  • Line 235: "While initial body weights did not differ, weaning weights on d 21 were significantly heavier in the 25-OH-D3 group (1.47 ± 0.24 kg) than in the Ctrl group (1.46 ± 0.24 kg; P = 0.046)." The numerical weights described are those at birth, where there are no differences. The ones that should be used are the weaning weights.

3.3. Effects of lactation stage and dietary 25-OH-D3 on the milk amino acid profile

  • The results presented in all tables are mean ± standard deviation or error?
  • Table 5. I don't believe the results for alanine, histidine, and lysine are significant. They shouldn't be significant, especially with an N of 10 sows per group.

3.4. Influence of lactation day and dietary 25-OH-D3 inclusion on the fatty acid profile of milk

  • Figure 4 and 5. There are two graphs for each parameter. These two represent the same thing. It could be simplified into a single graph: the graph on the right with the lines from the graph on the left. This way the graphics can be made bigger and they would look much better.

4. Discussion

  • Line 363. "In this study, sows supplemented with 25-OH-D3 did not change the litter size and weight, which align with others studies." In this study, supplementation was administered from day 85 of gestation, when embryo implantation had already occurred. Therefore, it is normal that there were no differences in the size and number of piglets born.
    When comparing with previously published articles, the timing of supplementation should be taken into account and mentioned. 
  • Line 446. "When compared to colostrum, the higher concentration of saturated fatty acids and lower content of polyunsaturated fatty acids were also noticed on 21 d of lactation." Rephrase the sentence. I understand that you want to compare the colostrum results from the literature with the results obtained with milk.
  • I have all the information from line 490 to line 513 missing. I don't think it contributes anything to the study. At least it should be summarized.

5. Conclusions

  • Linea 535. "reducing backfat loss and tear stains during lactation". Considering the methodology described, tear stains are not measured at any point during lactation.
    I have previously mentioned that there are inconsistencies between the methodology described and the results presented.

Author Response

Dear Editors and Reviewers:

We are truly grateful to yours and other reviewers’ critical comments and thoughtful suggestions. Based on these comments and suggestions we have made careful modifications on the original manuscript. All changes made to the text are in red color. We hope the new manuscript will meet your journal’s standard. Below you will find our point-by-point responses to the reviewers’ comments/ questions:

Comments and Suggestions for Authors

This study evaluates the effects of maternal 25-hydroxycholecalciferol (25-OH-D3) supplementation on sow, from day 85 of gestation until they are weaned. They analyze the performance of sows, the growth of offspring, and the amino acid and fatty acid profile of the milk.

Thank a lot for comments.

2. Materials and Methods

2.1. Animals, diets and management

Line 112: "at 85 d of pregnancy were allocated into two groups (10 sows/group)" The expression isn't correct. The sows aren't pregnant, they're gestating.

This true, we revised the description as following: “Using a randomized block design, a total of 20 primiparous Duroc × (Landrace × Large White) gestating sows were allocated into two groups (10 sows per group) at 85 d of gestation based on initial body weight and backfat thickness”. Thanks.

Line 115: "Dietary treatments were administered throughout gestation and lactation, with weaning performed on d 21 of lactation." The treatment is not during the entire gestation; it starts from day 85.

According to your suggestion, we revised the sentence as following: “Dietary treatments were performed from 85 d of pregnancy to on d 21 of lactation”

2.2. Back-fat thickness and sow body condition score

Line 139: "Two points located 65 mm from the dorsal midline, aligned with the last rib, were shaved on standing sows". Bad translation. Back-fat is not measured at two points, but at a single point called P2.

Thank you for suggestion, we revised the description as following:

“Back fat was measured at the P2 site using a digital backfat meter (Renco Lean-Meter, Minneapolis, USA) [24], in which the point of measurement was marked on each sow to guarantee that exactly the same place was investigated during the subsequent measurements”.

2.4. Performance of sows

I'm missing information on how food is administered to sows, both during gestation and lactation. How often do they eat? What are the feeders like? How is their intake measured? Therefore, how is each sow's intake calculated and recorded?

As for the referee’s concern, in this study, the sows were fed a basal diet (3,200 IU/kg vitamin D3) containing either 0 μg/kg (Ctrl group) or 50 μg/kg 25-OH-D3. Dietary treatments were performed from 85 d of pregnancy to on d 21 of lactation. Sows were maintained in individual stalls (2.13 × 0.61 m) for the duration of gestation (d 85-110) and provided a gestation diet at a level of 2.70 kg per day, divided into two feedings. On d 110 of gestation, sows were moved to farrowing pens and offered a lactation diet ad libitum. Feed intake was recorded via electronic feeders, which measured refusals to calculate daily consumption.

We added the related description in M&M section. Thanks.

Line 165: "Litter birth weight was measured, and individual piglet birth weights were calculated by dividing total litter weight by the number of live-born piglets". I understand that the litter weight, and therefore the individual weight of the piglets, is the weight of each sow once she has farrowed. After this, are piglets moved between sows? If so, birth weight is one thing, and post-movement weight is another. This latter weight should be taken into account when calculating growth and weight gain.

Herein, the litter birth weight was measured, and individual piglet birth weights were calculated by dividing total litter weight by the number of live-born piglets. However, after farrow, litter size was moved between sows within each group to ensure the survival rate and weaning weight of piglets. The adjusted litter weight and size were used to calculate the individual and litter weight gain. Therefore, we specified this information in the section of “2.4. Performance of sows”. Thanks.

3. Results

3.1. Tear stain score and body condition

Figure 1. The dates described in the methodology and those represented in the results do not match. For example, in methods back-fat thickness was measured on d 93, 101, and 108 of gestation, as well as d 20 of lactation. In figure there are d 85, delivery and d 20 of lactation. nothing to do with that.

We sorry for the confused description, in this study, back-fat thickness was measured by a single researcher on d 85 of pregnancy, delivery, and d 20 of lactation. However, the score of both body condition and tear stain were recorded on d 93, 101, and 108 of gestation.

3.2. Sow performance

Line 235: "While initial body weights did not differ, weaning weights on d 21 were significantly heavier in the 25-OH-D3 group (1.47 ± 0.24 kg) than in the Ctrl group (1.46 ± 0.24 kg; P = 0.046)." The numerical weights described are those at birth, where there are no differences. The ones that should be used are the weaning weights.

Indeed, we revised the sentence as following: “While initial body weights did not differ, weaning weights on d 21 were significantly heavier in the 25-OH-D3 group (6.16±0.48 kg) than in the Ctrl group (5.50±0.77 kg; P = 0.031)”.

3.3. Effects of lactation stage and dietary 25-OH-D3 on the milk amino acid profile. The results presented in all tables are mean ± standard deviation or error?

Thanks. Results are expressed as means and their standard errors.

Table 5. I don't believe the results for alanine, histidine, and lysine are significant. They shouldn't be significant, especially with an N of 10 sows per group.

As for the referee’s concern, we very careful checked the raw data and statistic them, the P-value did less than 0.05, if possible, we will share with you the raw data. Thanks.

3.4. Influence of lactation day and dietary 25-OH-D3 inclusion on the fatty acid profile of milk Figure 4 and 5. There are two graphs for each parameter. These two represent the same thing. It could be simplified into a single graph: the graph on the right with the lines from the graph on the left. This way the graphics can be made bigger and they would look much better.

In Figure 4 and 5, we want to express the alterations of fatty acids response to day of lactation and 25-OH-D3 treatment. For this, we separated Figure 5 to better display this alteration according to your suggestion.

4. Discussion

Line 363. "In this study, sows supplemented with 25-OH-D3 did not change the litter size and weight, which align with others studies." In this study, supplementation was administered from day 85 of gestation, when embryo implantation had already occurred. Therefore, it is normal that there were no differences in the size and number of piglets born. When comparing with previously published articles, the timing of supplementation should be taken into account and mentioned.

Thank your suggestion, we improved the sentence as following:

“Sows supplemented with 25-OH-D3 did not change the litter size and weight, which align with others studies. For example, evidence from Long et al. (2024) found that 50 µg/kg 25-OH-D3 supplementation had no significant effects on the number of piglets alive, stillborn rate, and litter weight [8]. Of note, this supplementation was administered from d 85 of gestation, when embryo implantation had already occurred. Therefore, it is normal that there were no differences in the size and number of piglets born”.

Line 446. "When compared to colostrum, the higher concentration of saturated fatty acids and lower content of polyunsaturated fatty acids were also noticed on 21 d of lactation." Rephrase the sentence. I understand that you want to compare the colostrum results from the literature with the results obtained with milk.

We specified the description as following:

“In this study, the analysis of the milk fatty acid profile showed that saturated fatty acids increased linearly and polyunsaturated fatty acids decreased linearly with advancing lactation, whereas the alterations of total fatty acids and monounsaturated fatty acids were unaffected. Similar patterns have also been noticed in previous study [23]”.

I have all the information from line 490 to line 513 missing. I don't think it contributes anything to the study. At least it should be summarized.

According to your suggestion, we deleted the paragraph.

5. Conclusions

Linea 535. "Reducing backfat loss and tear stains during lactation". Considering the methodology described, tear stains are not measured at any point during lactation. I have previously mentioned that there are inconsistencies between the methodology described and the results presented.

Indeed, we improved the conclusion as following: “The finding of the current study indicated that dietary supplementation with 50 μg/kg 25-OH-D3 improved weaning weight and weight gain and kept the sows in better physical condition. In response to the lactation day, amino acid levels were highest in co-lostrum and then fell. This drop was faster in sows given 25-OH-D3, resulting in lower amino acid levels by the end of lactation. Over the lactation period, saturated fats in-creased while polyunsaturated fats decreased. The 25-OH-D3 supplement first lowered all fat levels, then later (l4 to 21 d of lactation) increased specific beneficial fats, especially oleic, linoleic acids, and arachidonic acid. These findings support the hypothesis that maternal supplementation of 25-OH-D3 in gestation and lactation diets could alter milk composition, which appeared to be connected to the improved piglet growth”.

Reviewer 2 Report

Comments and Suggestions for Authors

I have some suggestions for the authors in an enclosed archive.

Comments on the Quality of English Language

The language can be improved to better understanding of the results.

Author Response

Dear Editors and Reviewers:

We are truly grateful to yours and other reviewers’ critical comments and thoughtful suggestions. Based on these comments and suggestions we have made careful modifications on the original manuscript. All changes made to the text are in red color. We hope the new manuscript will meet your journal’s standard. Below you will find our point-by-point responses to the reviewers’ comments/ questions:

I have some suggestions for the authors in an enclosed archive. The language can be improved to better understanding of the results. Suggestions and comments for ANIMALS-3922411-peer-review-v1

Thanks for your comments. The manuscript has been checked by a native English speaker researcher, and the manuscript has been improved accordingly.

Page Line It says: Suggestion:

1 18 …given the present… …given the presence…

We revised the word.

1 22-23 …rate of descent… …rate of reduction…

We changed the “descent” into “reduction”

1 35 …after being inseminated… This is not relevant, so, I suggest to

eliminate it

We deleted the “after being inseminated”.

1 39 …no significant was found… …no significant difference was found…

We have done corresponding revision according to this comment.

1 43 …rate of descent… …rate of reduction…

We revised the word.

2 59 …necessitating… …requiring…

The “necessitating” was changed into “requiring”. Thanks.

2 75-76 …insufficiency was found to link… …insufficiency was found to be linked…

We have done corresponding revision according to your comment.

2 83 …performance of litter major relies… …performance of litter mainly relies…

We replaced the “major” with “mainly”.

3 96 …showed… shows… Kind of cacophony. Try to avoid it.

We have done corresponding revision in the manuscript according to this comment.

3 97 …A elevates fat content… …An elevated fat content…

We changed the “elevates” into “elevated”.

3 112 …after being inseminated… Why is this relevant if the sows were already at the 85 days of gestation? I suggest to eliminate this statement

Indeed, we deleted the related statement.

7 Figure 1 The figures C, D and E do no help to interpret the results. Y suggest to use a Table with these data.

Thanks, we have transferred Figure 1 into Table.

7 222 …No significant was also… No significant difference was also…

We revised the sentence.

7 223 The gestation day you are referring to may be the parturition or delivery day. I suggest to change to any of these concepts.

In this study, the gestation day was defined as the day from confirm pregnancy, thus we hope keep the current concept if possible.

7 229 …on food consumption… on feed intake… Feed is used in animals’ intake; food is usually used for human intake.

We replaced the “food” with “feed”.

8 Table 3 Gestation day Change to Parturition or Delivery day.

We hope keep the current concept, if possible, based on the concept of gestation day defined as the day from confirm pregnancy.

8 Figure 3 I suggest to use a Table. The figures A-D are too confusing to interpret the results

In this study, the data were presented in Table 6. Thanks.

9 247 …Data was analyzed… …Data were analyzed…

We changed “was” into “were”.

9 251 …following drop dramatically… …following a dramatic reduction…

We have done corresponding revision in the manuscript according to this comment.

9 252 …the rate of descent… …the rate of reduction…

We revised the word.

9 257 …resulted to decreased… …resulted in decreased…

We have done corresponding revision. Thanks.

10 Figure 3 All figures are too confusing to interpret the results. I suggest to use just the Table 5 to show the results.

The data of Figure 3 have been presented in Table 6 in this study.

13 Figure 4 All figures are too confusing to interpret the results. I suggest to use a Table to show the results.

In this study, the data of Figure 4 were presented in Table 7. Thanks.

13 288-9 …taken out lower… This statement is confusing. Please explain in a better way the results

We improved the descript as following: “The colostrum from sows supplemented with 25-OH-D3 exhibited a significant (P < 0.05) decrease in the content of palmitic, heptadecanoic, and stearic acids compared to the Ctrl group. These reductions were maintained until d 3 of lactation, with the decrease in stearic acid extending to d 7”.

14 Figure 5 All figures are too confusing to interpret the results. I suggest to use a Table to show the results.

The data of Figure 3 have been presented in Table 7 in this study.

18 389 …major relies… …mainly relies…

We revised the word.

18 400 …following drop dramatically… …followed by a dramatic reduction…

We have done corresponding revision.

18 400-1 …which highted the importance… …which highlighted the importance…

We improved the sentence as you suggestion.

19 432 …the precise underlying mechanism underling the milk… This is a confusing statement. Please try again.

The sentence was revised as following: “the precise mechanism by which milk amino acids influence litter performance needs further elucidation”.

20 475 …was observed to attenuates mucosal damage… …was observed to attenuate mucosal damage…

We revised the word.

20 480-1 …fatty acids myristic, palmitic, and stearic acid… …fatty acids myristic, palmitic and stearic…

Thanks. We have done corresponding revision.

20 481-2 …fatty acids oleic, linoleic, 11-eicosenoic, and arachidonic acid… …fatty acids oleic, linoleic, 11-eicosenoic and arachidonic…

We have done corresponding revision in the manuscript according to this comment.

20 486 …fatty acids, such as oleic, linoleic, arachidonic acid… …fatty acids, such as oleic, linoleic and arachidonic…

We added the “and” in this sentence.

20 490 …There was little information is available about… …There was little information available about…20 494 …by its regulation of key transcriptional regulators… Please try again to avoid cacophony.20 501 …fatty acid synthesis was confirmed to involve in the activation……fatty acid synthesis was confirmed to be involved in the activation…20 507 …vitamin D3 was found could regulate lipid metabolism… …vitamin D3 was found that could regulate lipid metabolism…20 511-2 …was also deemed to modulates fat synthesis… …was also deemed to modulate fat synthesis…20 516 …altering mammary gland lipogenic… …altering mammary gland lipogenic activity…

Thank your professional suggestion, we deleted the content according to other reviewers’ advice.

21 539-40 …following drop dramatically… …following by a dramatic reduction…

We improved the descript.

21 540 …The rate of descent… …The rate of reduction…

Thanks. We have done corresponding revision.

21 549 …especially oleic, linoleic acids, arachidonic acid… …especially oleic, linoleic

and arachidonic acids…

We added the “and” in this sentence.

Reviewer 3 Report

Comments and Suggestions for Authors

General Comment
The article entitled “Maternal 25-hydroxycholecalciferol supplementation dynamically altered milk fatty acid and amino acid profiles and improves sow performance” is a promising study, and the dataset is potentially valuable. However, numerical/chronological inconsistencies, missing units, statistical clarity, and data corrections need to be addressed to ensure robustness and reproducibility.

Abstract
The authors state that supplementation increased weaning weight, but no numerical values are provided. This omission is critical, as values must be explicitly reported. Highlighting such results is particularly important because the central impact (dynamic alterations in milk composition combined with piglet performance) could be emphasized more strongly, while some secondary details could be shortened.

Introduction
The authors present the topic appropriately, provide readers with relevant knowledge gaps, and physiologically justify the support for supplementation in the parameters analyzed. However, one aspect requires improvement. The introduction links maternal performance and litter outcomes, expands on the broader role of 25-OH-D3, and defines the objective of evaluating its effects on performance and dynamic milk composition. Nevertheless, in lines 69–79, the writing is somewhat confusing, as the authors describe possible effects of 25-hydroxycholecalciferol in other species, then conclude by stating that no notable effects were found on reproductive variables, and immediately afterward cite a positive result. The intention of highlighting divergent results in the literature is weakened by presenting only a brief description of contradictory findings. I recommend revising the text to include a concise analytical commentary that better justifies the gap identified in lines 78–90.

Materials and Methods
In Table 1, the row “Metabolic energy, kcal/kg” is left blank, which hinders interpretation of isoenergetic comparisons between treatments.
Two methods were used (ultrasound and caliper), but no concordance analysis between them is presented. What was the rationale for applying both methods—validation, methodological comparison, or control? This is not clearly stated.
In line 154, the authors refer to “gestation day,” and in line 155 they write “piglet,” which is confusing. The wording should be revised.
In line 172, sampling times are described as 24 h, d3, d5, d14, and d20, whereas in Tables 5 and 6, days 0, 3, 7, 14, and 21 of lactation are reported. The manuscript must be standardized regarding sampling days.
Section 2.7 (statistical analysis) contains inconsistencies that need clarification. The design involved repeated measures per sow across lactation days, but the text does not specify which covariance structure was used in GLIMMIX, nor whether “day” was included as a fixed effect with diet × day interaction. Furthermore, the mention of an independent t-test to compare the two treatments is potentially inappropriate for repeated measures and may inflate type I error. The scope of the Bonferroni correction is also unclear—whether applied per variable, per day, or per hypothesis family—and the number of comparisons corrected is not stated. This issue is particularly relevant given the large number of amino acids and fatty acids analyzed. It is strongly recommended to specify the model used in each analysis (performance, fatty acids, amino acids) and the approach to multiple testing, to ensure transparency and statistical robustness.

Results
In line 231, it is reported that after “standardization,” the Ctrl group showed higher averages of live-born piglets (12.9 vs 11.0; P<0.001) and weaned piglets (12.5 vs 10.3; P=0.001). However, the consequence of standardization is not clearly explained. Standardization usually equalizes the litter size per sow, yet here it resulted in distinct means between treatments. The method and rationale should be clarified.
In line 236, the authors erroneously report weaning weights, but the values provided correspond to birth weights. The correct weaning weights are presented in Table 4.
In Table 6, the units of measurement are not indicated, which compromises data interpretation.

Discussion
The authors provide a plausible mechanistic integration of their results, supported by appropriate references that help contextualize and interpret the findings.

Conclusion
The conclusion presents the relevant findings and answers the research questions. However, it should be written more concisely. As it stands, it gives the impression that the data are being re-discussed, which has already been adequately addressed in the Discussion section.

Author Response

Dear Editors and Reviewers:

We are truly grateful to yours and other reviewers’ critical comments and thoughtful suggestions. Based on these comments and suggestions we have made careful modifications on the original manuscript. All changes made to the text are in red color. We hope the new manuscript will meet your journal’s standard. Below you will find our point-by-point responses to the reviewers’ comments/ questions:

The article entitled “Maternal 25-hydroxycholecalciferol supplementation dynamically altered milk fatty acid and amino acid profiles and improves sow performance” is a promising study, and the dataset is potentially valuable. However, numerical/chronological inconsistencies, missing units, statistical clarity, and data corrections need to be addressed to ensure robustness and reproducibility.

We appreciate the reviewer’s comments, and have done corresponding revision in whole manuscript according to your comment.

Abstract

The authors state that supplementation increased weaning weight, but no numerical values are provided. This omission is critical, as values must be explicitly reported. Highlighting such results is particularly important because the central impact (dynamic alterations in milk composition combined with piglet performance) could be emphasized more strongly, while some secondary details could be shortened.

As for the referee’s concern, we added the numbers of increased weaning weight in Abstract section as following: “whereas maternal 25-OH-D3 intervention notably increased weaning weight and weight gain of piglet (P < 0.05), in which the dietary 25-OH-D3 supplementation contributed to 16.4% body gain during lactation”. Thanks.

Introduction

The authors present the topic appropriately, provide readers with relevant knowledge gaps, and physiologically justify the support for supplementation in the parameters analyzed. However, one aspect requires improvement. The introduction links maternal performance and litter outcomes, expands on the broader role of 25-OH-D3, and defines the objective of evaluating its effects on performance and dynamic milk composition. Nevertheless, in lines 69–79, the writing is somewhat confusing, as the authors describe possible effects of 25-hydroxycholecalciferol in other species, then conclude by stating that no notable effects were found on reproductive variables, and immediately afterward cite a positive result. The intention of highlighting divergent results in the literature is weakened by presenting only a brief description of contradictory findings. I recommend revising the text to include a concise analytical commentary that better justifies the gap identified in lines 78–90.

According to your professional suggestion, we improved these description as following:

“Clinical data showed that women with reproductive dysfunction had decreased serum levels of 25-OH-D3 compared to normal individuals [5, 7]. In the pig breeding industry, the dietary supplementation of 25-OH-D3 at 50 or 200 µg/kg to sows had no significant effect on the number of piglets born alive, litter weight, or suckling piglet performance [8, 9]. However, maternal 25-OH-D3 supplementation at a dosage of 50 μg/kg has been reported to increase weaning litter weight and total litter weight gain [10, 11]. Potential factors con-tributing to these results include the total dosage of vitamin D3, differences in feeding management, the duration of the trial, and the parity of the sows. Maternal vitamin D in-sufficiency was found to link with decreased fertility and adverse pregnancy outcomes [12], which also accompanied by a reduction in breast milk vitamin D content, offspring birth weight, and neonatal bone mineral content [13]. Supplementing sow diets with 50 μg/kg of 25-OH-D3 has been shown to significantly increase weaning litter weight, with reported gains of 10.0% [11] and up to 19.7% [10]. These findings suggest that supplementing sows with 25-OH-D3 during gestation and lactation may ensure adequate ma-ternal vitamin D status, thereby improving sow performance and offspring development”.

Materials and Methods

In Table 1, the row “Metabolic energy, kcal/kg” is left blank, which hinders interpretation of isoenergetic comparisons between treatments.

Thanks. We provide the value of Metabolic energy in Table 1.

Two methods were used (ultrasound and caliper), but no concordance analysis between them is presented. What was the rationale for applying both methods—validation, methodological comparison, or control? This is not clearly stated.

In this study, backfat thickness was measured using a caliper and digital backfat meter, both the two methods usually are used in most of pig farms in China. Although some evidence showed there is a high correlation between them, using the two methods might be more accurate for evaluating the effects of 25-OH-D3 on backfat thickness.

In line 154, the authors refer to “gestation day,” and in line 155 they write “piglet,” which is confusing. The wording should be revised.

We revised the word.

In line 172, sampling times are described as 24 h, d3, d5, d14, and d20, whereas in Tables 5 and 6, days 0, 3, 7, 14, and 21 of lactation are reported. The manuscript must be standardized regarding sampling days.

We are very sorry for this confusion, this is a clerical error, it should be d 1, 3, 7, 14, and 21 of lactation.

Section 2.7 (statistical analysis) contains inconsistencies that need clarification. The design involved repeated measures per sow across lactation days, but the text does not specify which covariance structure was used in GLIMMIX, nor whether “day” was included as a fixed effect with diet × day interaction. Furthermore, the mention of an independent t-test to compare the two treatments is potentially inappropriate for repeated measures and may inflate type I error. The scope of the Bonferroni correction is also unclear—whether applied per variable, per day, or per hypothesis family—and the number of comparisons corrected is not stated. This issue is particularly relevant given the large number of amino acids and fatty acids analyzed. It is strongly recommended to specify the model used in each analysis (performance, fatty acids, amino acids) and the approach to multiple testing, to ensure transparency and statistical robustness.

I very appreciate your good advice. We have done corresponding revision according to this comment, as following:

Data was analyzed using the GLIMMIX procedure in SAS (SAS Institute, Inc., Cary, NC) and considered sow (litter) as the study unit. The data obtained were analyzed by the Shapiro-Wilk and Levene’s test to assess normal distribution and homogeneity of variances. An independent t-test was used to compare Significant differences between the diet with and without 25-OH-D3 groups on sow performance, fatty acids, and amino acids. A Bonferroni correction was applied to control for multiple comparisons. One-way Analysis of Variance (ANOVA) followed by Tukey’s test for multiple comparisons was performed to elucidate the effect of lactation day on the fatty acid and amino acid profiles. The statistical model applied was:

Yi= μ +Dii

Where is the response variable, μ is the overall mean,  Di is the fixed effect of dietary 25-OH-D3 or day of lactation, and ξi is the error term.

In addition, the performance of sows during lactation between Ctrl and 25-OH-D3 groups was compared using one-way analysis of covariance (ANCOVA). Adjusted litter size was included as covariates in a single ANCOVA model.

Results

In line 231, it is reported that after “standardization,” the Ctrl group showed higher averages of live-born piglets (12.9 vs 11.0; P<0.001) and weaned piglets (12.5 vs 10.3; P=0.001). However, the consequence of standardization is not clearly explained. Standardization usually equalizes the litter size per sow, yet here it resulted in distinct means between treatments. The method and rationale should be clarified.

Herein, the litter birth weight was measured, and individual piglet birth weights were calculated by dividing total litter weight by the number of live-born piglets. However, after farrow, litter size was moved between sows within each group to ensure the survival rate and weaning weight of piglets. The adjusted litter weight and size were used to calculate the individual and litter weight gain. Therefore, we specified this information in the section of “2.4. Performance of sows”. Thanks.

In line 236, the authors erroneously report weaning weights, but the values provided correspond to birth weights. The correct weaning weights are presented in Table 4.

Sorry for this mistake, we revised the sentence as following: “While initial body weights did not differ, weaning weights on d 21 were significantly heavier in the 25-OH-D3 group (6.16±0.48 kg) than in the Ctrl group (5.50±0.77 kg; P = 0.031)”.

In Table 6, the units of measurement are not indicated, which compromises data interpretation.

We added the units of fatty acids. Thanks.

Discussion

The authors provide a plausible mechanistic integration of their results, supported by appropriate references that help contextualize and interpret the findings.

Thank you for comments.

Conclusion

The conclusion presents the relevant findings and answers the research questions. However, it should be written more concisely. As it stands, it gives the impression that the data are being re-discussed, which has already been adequately addressed in the Discussion section.

According to you suggests, we simplified the conclusion and made more concisely, as following:

“The finding of the current study indicated that dietary supplementation with 50 μg/kg 25-OH-D3 improved weaning weight and weight gain and kept the sows in better physical condition. In response to the lactation day, amino acid levels were highest in colostrum and then fell. This drop was faster in sows given 25-OH-D3, resulting in lower amino acid levels by the end of lactation. Over the lactation period, saturated fats increased while polyunsaturated fats decreased. The 25-OH-D3 supplement first lowered all fat levels, then later (l4 to 21 d of lactation) increased specific beneficial fats, especially oleic, linoleic acids, and arachidonic acid. These findings support the hypothesis that maternal supplementation of 25-OH-D3 in gestation and lactation diets could alter milk composition, which appeared to be connected to the improved piglet growth”.

Thanks.

Reviewer 4 Report

Comments and Suggestions for Authors

This study evaluated the effects of dietary 25-hydroxycholecalciferol (25-OH-D3) supplementation to sows during gestation and lactation on sow performance and milk composition. Results showed that 25-OH-D3 inclusion increased piglet weaning weight and daily weight gain, improved milk fatty acid and amino acid profiles, and specifically elevated functional fatty acids such as oleic, linoleic and arachidonic acids during mid-to-late lactation, thereby enhancing piglet growth performance and sow health status. However, the manuscript itself contains several deficiencies that need revision.

1 There are multiple grammatical errors in the manuscript; the authors are advised to check the entire text carefully. A few simple examples are given below.

(1) "It was well-known that nutritional interventions during gestation can alter organ structure and influence prenatal and neonatal growth, as well as weight gains in newborn pigs [3]." “It was well-known” is an informal expression that lacks academic rigor. 

(2) "Clinic data showed the women with reproductive dysfunction exhibited decreased serum contents of …. individual [5, 7]." “Clinic data” should be “Clinical data”; “individual” should be plural; “the women” is redundant. 

(3) "As far as pig is concerned, there were no notable effects on the number of piglets alive, litter weight, … [8, 9]. "The sentence structure is loose and fails to clearly distinguish differences between study findings. 

(4) "Interestingly, the vitamin …, was presented in mammary gland [20], implied that maternal …composition." “was presented” should be “are expressed”; “implied” should be “implying”. 

(5) "However, the data highlight the effects … composition of amino acid and fatty acids still requires further elaboration." The sentence structure is chaotic and lacks subject consistency.

2 L64-65 The phrase "bypasses liver metabolism" is inaccurate because 25-OH-D3 itself is produced in the liver.

3 L71-75 Please clearly identify the potential reasons for the inconsistency in the existing research results.

4 L81-83 Please additionally clarify how specific amino acids and fatty acids in milk concretely affect piglet development.

5 L92-95 Please further elaborate on the potential regulatory mechanisms of VDR in the mammary gland.

6 L111-113 Please indicate whether randomization was applied to the allocation of treatments and whether baseline parameters such as initial body weight and backfat thickness were taken into account.

7 L113-114 Please clarify how many IU of vitamin D₃ 50 µg/kg 25-OH-D₃ corresponds to, the total vitamin D activity equivalent, and the rationale for selecting this dose.

8 L140-146 The same trait was measured with two devices; state why both methods were employed and furnish the concordance between them.

9 L37&171-174 Why are two different sampling schedules (d 1, 3, 7, 14, 21 vs d 1, 3, 5, 14, 20) presented in the text, which contradict each other?

10 L197-200 Why was lactation day not included as a fixed effect in the model, given that the present study focuses on dynamic changes? The description of random effect δj is unclear; please specify what "study unit" refers to.

11 L231-233 A significant difference existed between the two groups in the key baseline indicator of adjusted live-born piglet count. This could heavily influence the interpretation of subsequent weaning weight and weight gain, as the number of suckling piglets differed. However, the authors did not include this as a covariate when comparing weaning performance. It is recommended that initial litter size be used as a covariate for statistical adjustment (ANCOVA) when comparing weaning weight and weight gain, or that the potential limitation of this baseline difference on result interpretation be explicitly discussed in the results.

12 L268 In Table 5, parameters of a one-phase decay model such as “Plateau” and “K (rate of descent)” were introduced abruptly. The statistical methods section did not specify that a one-phase decay model would be used to fit the temporal trends of amino-acid data, and the Results section presented these model parameters without explaining the model-selection criteria or the meaning of the goodness-of-fit (R²).

13 L270 Figure 4 covers total fatty acids, SFA, MUFA, PUFA and other indices, yet the main text refers to the whole of Figure 4 without precision. Moreover, it claims that MUFA showed "very little fluctuation" and provides a P value, but does not indicate what test this P value comes from (time effect or treatment effect?).

14 L322-324 The observed performance improvement "may be attributed to" changes in milk composition; however, this study employed a correlational design and cannot establish causation. Other unmeasured factors (e.g., piglet gut health, milk intake) could also contribute. A more cautious phrasing should treat milk composition as an associated factor rather than a cause.

15 L332-334 The authors infer reduced sow stress from decreased tear-stain scores. However, compared with rodents, the validation of tear-stain scoring in pigs and its precise association with stress remain insufficient; drawing a "stress-reduction" conclusion based solely on this single indicator appears weak.

16 L378-383 The authors attribute the observed growth improvement to the IGF-1 increase reported in other studies [21, 35]. However, the present study did not measure IGF-1 or any related hormone levels. Consequently, this mechanistic explanation relies entirely on external evidence and has only weak direct relevance to the current trial. The authors should explicitly state that this mechanism is speculative, derived from the literature, and acknowledge this as a limitation of their study.

17 L393-396 The reasoning assumes that: 1) feed intake and backfat loss are perfect proxies for milk yield; 2) milk yield was indeed identical between the two groups. Yet, differences in milk-ejection efficiency or mammary metabolic efficiency could also result in dissimilar milk output or composition under similar feed intake and body-tissue mobilization. The conclusion “likely attributable” is overly assertive.

18 L427-429 The line of reasoning assumes that because the authors observed declines in the plateau concentrations of several amino acids yet detected no differences in leucine, threonine, or valine, these alterations are irrelevant to the improved growth. Such a conclusion overlooks the complexity of the overall amino acid profile, its balance, and interactive effects. The absence of change in a single amino acid does not exclude the collective impact of changes in others; absolute dismissal should be avoided, and the potential influence of the overall pattern should instead be discussed.

19 The authors described in detail how 25-OH-D₃ initially decreased fatty acids and later elevated specific fatty acids (e.g., oleic, linoleic, arachidonic acid). However, the discussion did not elaborate on the physiological significance of this precise temporal dynamic for piglets.

20 L490-518 This paragraph extensively cites putative roles of VDR, SREBP1, INSIG1, ACC and FAS in mammary and non-mammary tissues; however, none of these genes or proteins were quantified in the present study. The discourse is largely literature-driven speculation, loosely connected to our empirical data and should be shortened and explicitly framed as a hypothesis awaiting verification.

Comments on the Quality of English Language

Please refer to Reviewer Comment 1.

Author Response

Dear Editors and Reviewers:

We are truly grateful to yours and other reviewers’ critical comments and thoughtful suggestions. Based on these comments and suggestions we have made careful modifications on the original manuscript. All changes made to the text are in red color. We hope the new manuscript will meet your journal’s standard. Below you will find our point-by-point responses to the reviewers’ comments/ questions:

This study evaluated the effects of dietary 25-hydroxycholecalciferol (25-OH-D3) supplementation to sows during gestation and lactation on sow performance and milk composition. Results showed that 25-OH-D3 inclusion increased piglet weaning weight and daily weight gain, improved milk fatty acid and amino acid profiles, and specifically elevated functional fatty acids such as oleic, linoleic and arachidonic acids during mid-to-late lactation, thereby enhancing piglet growth performance and sow health status. However, the manuscript itself contains several deficiencies that need revision.

Thanks for comments.

 1 There are multiple grammatical errors in the manuscript; the authors are advised to check the entire text carefully. A few simple examples are given below.

The manuscript has been checked by a native English speaker researcher, and the manuscript has been improved accordingly. Thanks.

(1) "It was well-known that nutritional interventions during gestation can alter organ structure and influence prenatal and neonatal growth, as well as weight gains in newborn pigs [3]." “It was well-known” is an informal expression that lacks academic rigor.

We revised the sentence as follows:

“Multiple evidence indicated that nutritional interventions during gestation can alter organ structure and influence prenatal and neonatal growth”.

(2) "Clinic data showed the women with reproductive dysfunction exhibited decreased serum contents of …. individual [5, 7]." “Clinic data” should be “Clinical data”; “individual” should be plural; “the women” is redundant.

Thanks. We improved the expression as following: “Clinical data showed that women with reproductive dysfunction had decreased serum levels of 25-OH-D3 compared to normal individuals”.

(3) "As far as pig is concerned, there were no notable effects on the number of piglets alive, litter weight, … [8, 9]. "The sentence structure is loose and fails to clearly distinguish differences between study findings.

According to your suggestion, the sentence structure has been improved as following: “In the pig breeding industry, the dietary supplementation of 25-OH-D3 at 50 or 200 µg/kg to sows had no significant effect on the number of piglets born alive, litter weight, or suckling piglet performance [8, 9]”.

(4) "Interestingly, the vitamin …, was presented in mammary gland [20], implied that maternal …composition." “was presented” should be “are expressed”; “implied” should be “implying”.

We have done corresponding revision according to this comment.

(5) "However, the data highlight the effects … composition of amino acid and fatty acids still requires further elaboration." The sentence structure is chaotic and lacks subject consistency.

Thank you for suggestions, we revised the sentence as following: “Nonetheless, the effects of maternal 25-OH-D3 inclusion on milk composition of amino acids and fatty acids require further elaboration”.

2 L64-65 The phrase "bypasses liver metabolism" is inaccurate because 25-OH-D3 itself is produced in the liver.

Indeed, we revised the sentence as following: “25-hydroxycholecalciferol (25-OH-D3) is the major circulating metabolite of vitamin D, produced in the liver”.

3 L71-75 Please clearly identify the potential reasons for the inconsistency in the existing research results.

We added the probably explain as follows: “Potential factors contributing to these results include the total dosage of vitamin D3, differences in feeding management, the duration of the trial, and the parity of the sows”.

4 L81-83 Please additionally clarify how specific amino acids and fatty acids in milk concretely affect piglet development.

Thank you for your suggestions. We provided some evidence that highlights how specific amino acids and fatty acids in milk concretely affect piglet development, as follows:

Suckling pig growth depends on the availability and composition of milk, as well as the efficient conversion of its nutrients into body weight, suggesting that the growth performance of litter major relies on the amount and quality of colostrum and milk [14]. For example, threonine was linked to the production of immuno-globulin G (IgG) and played important roles in neonatal survival and infection prevention [15]. The supplementation of leucine is critical for intestinal development of suckling pig through enhancing muscle protein synthesis and gut maturation [16]. Some specific fatty acids were also observed to exert an indispensable role in the development of piglets such as oleic and linoleic acids. The improvement in piglet growth and survivability associated with dietary oleic and linoleic acid supplementation is attributed to their role as preferred energy substrates [17], their anti-inflammatory properties [18], and their ability to enhance mucosal development [19].

5 L92-95 Please further elaborate on the potential regulatory mechanisms of VDR in the mammary gland.

According to your advice, we added some potential regulatory mechanisms of VDR in the mammary gland, as following:

“Interestingly, the vitamin D receptor and 1α-hydroxylase, a key enzyme responsible for converting vitamin D into its active form (1,25-(OH)2-D3), are presented in mammary gland [20]. Evidence confirmed that the expression of sterol regulatory element-binding protein 1 (SREBP1) and its downstream target enzymes that mediate fatty acid synthesis were involved in the activation of vitamin D receptor [21]. Dietary 25-OH-D3 intervention promoted the expressions of acetyl-CoA carboxylase (ACC) and fatty acid synthase (FAS) in the breast tissue of lactating sows [22]. These findings imply that maternal 25-OH-D3 intervention could modify the milk composition”.

6 L111-113 Please indicate whether randomization was applied to the allocation of treatments and whether baseline parameters such as initial body weight and backfat thickness were taken into account.

In practice, a randomized block design was used in this study. According to Figure 1 and Table 3, there were no differences in terms of initial body weight and backfat thickness of sow. To clear the design, we specified the description as follows: “Using a randomized block design, a total of 20 primiparous Duroc × (Landrace × Large White) sows were allocated into two groups (10 sows per group) at 85 days of gestation based on initial body weight and backfat thickness. All sows had been inseminated twice with semen from the same Landrace boar”.

7 L113-114 Please clarify how many IU of vitamin D₃ 50 µg/kg 25-OH-D₃ corresponds to, the total vitamin D activity equivalent, and the rationale for selecting this dose.

Thanks. The dose of vitamin has been clarified as follows: “The sows were fed a basal diet (3,200 IU/kg vitamin D3) containing 25-OH-D3 at either 0 μg/kg (Ctrl group) or 50 μg/kg (corresponding to 2000 IU/kg vitamin D3), according to previous study [10]”.

8 L140-146 The same trait was measured with two devices; state why both methods were employed and furnish the concordance between them.

In this study, backfat thickness was measured using a caliper and digital backfat meter, both the two methods usually are used in most of pig farms in China. Although some evidence showed there is a high correlation between them, using the two methods might be more accurate for evaluating the effects of 25-OH-D3 on backfat thickness.

9 L37&171-174 Why are two different sampling schedules (d 1, 3, 7, 14, 21 vs d 1, 3, 5, 14, 20) presented in the text, which contradict each other?

We are very sorry for this confusion, this is a clerical error, it should be d 1, 3, 7, 14, and 21 of lactation.

10 L197-200 Why was lactation day not included as a fixed effect in the model, given that the present study focuses on dynamic changes? The description of random effect δj is unclear; please specify what "study unit" refers to.

For the referee’s concern, the sow (litter) was considered as the study unit. We used to analyze the effects of 25-OH-D3 and day of lactation on targeted parameters, and showed that there no interaction between 25-OH-D3 and day of lactation. Therefore, we just analyzed the main effects.

We realized that the description was unclear. Thus, we improved the Statistical analysis as following:

“Data was analyzed using the GLIMMIX procedure in SAS (SAS Institute, Inc., Cary, NC) and considered sow (litter) as the study unit. The data obtained were analyzed by the Shapiro-Wilk and Levene’s test to assess normal distribution and homogeneity of variances. Significant differences between the diet with and without 25-OH-D3 groups were analyzed by an independent sample t-test procedure with Bonferroni correction. One-way Analysis of Variance (ANOVA) followed by Tukey’s test for multiple comparisons was performed to elucidate a potential response by day of lactation. The statistical model applied was:

Yi= μ +Dii

WhereYi is the response variable, μ is the overall mean, Di is the fixed effect of dietary 25-OH-D3 or day of lactation, and εi is the error term.

11 L231-233 A significant difference existed between the two groups in the key baseline indicator of adjusted live-born piglet count. This could heavily influence the interpretation of subsequent weaning weight and weight gain, as the number of suckling piglets differed. However, the authors did not include this as a covariate when comparing weaning performance. It is recommended that initial litter size be used as a covariate for statistical adjustment (ANCOVA) when comparing weaning weight and weight gain, or that the potential limitation of this baseline difference on result interpretation be explicitly discussed in the results.

According to your professional suggestion, we compared the performance of sows during lactation between Ctrl and 25-OH-D3 groups using one-way analysis of covariance (ANCOVA). Adjusted litter size was included as covariates in a single ANCOVA model. And the results showed in Table 4 and was descripted as following: “Therefore, ANCOVA was performed with the adjusted litter size serving as covariates in a single ANCOVA model. The analysis showed that the number of weaned piglets and survivability to weaning were similar between groups (Table 5).”

12 L268 In Table 5, parameters of a one-phase decay model such as “Plateau” and “K (rate of descent)” were introduced abruptly. The statistical methods section did not specify that a one-phase decay model would be used to fit the temporal trends of amino-acid data, and the Results section presented these model parameters without explaining the model-selection criteria or the meaning of the goodness-of-fit (R²).

Thanks. We provided the illustration as following:

The model of one phase decay was present as following: Y=(Yo-Plateau) *exp(-K*Day) + Plateau

Where Yo is the Y value when Day is 1 d of lactation, Plateau is the Y value at infinite times, and K represent the rate of reduction.

13 L270 Figure 4 covers total fatty acids, SFA, MUFA, PUFA and other indices, yet the main text refers to the whole of Figure 4 without precision. Moreover, it claims that MUFA showed "very little fluctuation" and provides a P value, but does not indicate what test this P value comes from (time effect or treatment effect?).

Thanks. These alterations just aimed at the Ctrl group; therefore, we specified the content in the revised manuscript.

14 L322-324 The observed performance improvement "may be attributed to" changes in milk composition; however, this study employed a correlational design and cannot establish causation. Other unmeasured factors (e.g., piglet gut health, milk intake) could also contribute. A more cautious phrasing should treat milk composition as an associated factor rather than a cause.

Indeed. The improved weaning weights of piglets was associated with the alterations in milk composition as an associated factor rather than a cause. For this, we have done corresponding revision in the manuscript according to this comment.

15 L332-334 The authors infer reduced sow stress from decreased tear-stain scores. However, compared with rodents, the validation of tear-stain scoring in pigs and its precise association with stress remain insufficient; drawing a "stress-reduction" conclusion based solely on this single indicator appears weak.

There indeed lack strong evidence to support the conclusion, therefore, we revised the sentence as following:

 “It is therefore possible that dietary 25-OH-D3 supplementation might alleviate sow stress throughout farrowing and lactation. However, more convincing evidences are required to provide for supporting this concept in the present study.”

16 L378-383 The authors attribute the observed growth improvement to the IGF-1 increase reported in other studies [21, 35]. However, the present study did not measure IGF-1 or any related hormone levels. Consequently, this mechanistic explanation relies entirely on external evidence and has only weak direct relevance to the current trial. The authors should explicitly state that this mechanism is speculative, derived from the literature, and acknowledge this as a limitation of their study.

Thanks. We have done corresponding revision in the manuscript according to this comment, as following: “Based on these external evidence, it is possible that the growth enhancement observed in piglets from sows supplemented with 25-OH-D3 might be due to the activation of the IGF-1 axis”.

And we pointed out the limitation as following:

“Furthermore, the study lacked data on how 25-OH-D3 manipulation affects key components of the IGF-1 axis, including IGF-1 and GH”.

17 L393-396 The reasoning assumes that: 1) feed intake and backfat loss are perfect proxies for milk yield; 2) milk yield was indeed identical between the two groups. Yet, differences in milk-ejection efficiency or mammary metabolic efficiency could also result in dissimilar milk output or composition under similar feed intake and body-tissue mobilization. The conclusion “likely attributable” is overly assertive.

Thanks for your suggestions, we redefined the inclusion as following:

“Therefore, the undifferentiated feed intake and backfat loss of sows between Ctrl sows and 25-OH-D3 sows during lactation may, to some extent, reflected similar amount of milk production, implying that the milk quality might be related to the enhancements in litter and piglet weight gain during lactation in the current study”.

18 L427-429 The line of reasoning assumes that because the authors observed declines in the plateau concentrations of several amino acids yet detected no differences in leucine, threonine, or valine, these alterations are irrelevant to the improved growth. Such a conclusion overlooks the complexity of the overall amino acid profile, its balance, and interactive effects. The absence of change in a single amino acid does not exclude the collective impact of changes in others; absolute dismissal should be avoided, and the potential influence of the overall pattern should instead be discussed.

Thanks so much for your suggestions. We really overlooked the complexity of the overall amino acid profile, its balance, and interactive effects. For corrected this, we revised the sentence as following:

“Nevertheless, the comparable concentrations of leucine, threonine, and valine in the Ctrl and 25-OH-D3 groups do not rule out their contribution to the enhanced litter weight gain. The complex balance and interactive effects within the overall amino acid profile mean their role cannot be definitively assessed without further study”.

19 The authors described in detail how 25-OH-D₃ initially decreased fatty acids and later elevated specific fatty acids (e.g., oleic, linoleic, arachidonic acid). However, the discussion did not elaborate on the physiological significance of this precise temporal dynamic for piglets.

Thanks, we added the related statement as following:

“It was established that the precise temporal dynamics of fatty acids in piglets constitute a sophisticated evolutionary adaptation, ensuring the right fuel and structural components are available to overcome sequential physiological hurdles like the neonatal energy-thermal crisis, the development of the brain and immune system, and the challenges of weaning. In this study, the initial shift in the fatty acid ratio and the subsequent increase in beneficial long-chain unsaturated fatty acids, such as oleic, linoleic, arachidonic acid, likely provides piglets with a superior energy source and essential building blocks for organ development”.

20 L490-518 This paragraph extensively cites putative roles of VDR, SREBP1, INSIG1, ACC and FAS in mammary and non-mammary tissues; however, none of these genes or proteins were quantified in the present study. The discourse is largely literature-driven speculation, loosely connected to our empirical data and should be shortened and explicitly framed as a hypothesis awaiting verification.

Indeed, we removed the related content and shortened them. Moreover, the statement might be one of limitations due to the related data lacking, thus we revised these statements as following:

“Genes, proteins related to both amino acids and glycolipid metabolism and related rate-limiting enzymes should be determined to further define the alteration in milk fatty acid or amino acid profile due to dietary 25-OH-D3 supplementation, which was not performed in our study. Furthermore, the study lacked data on how 25-OH-D3 manipulation affects key components of the IGF-1 axis, including IGF-1 and GH. This omission prevents a mechanistic understanding of how dietary 25-OH-D3 supplementation alters milk composition and sow performance”.

Round 2

Reviewer 1 Report

Comments and Suggestions for Authors

No further comments

Reviewer 3 Report

Comments and Suggestions for Authors

After a further reading of the manuscript and the responses to the reviewers’ recommendations, I confirm that the suggested revisions have been properly incorporated and the necessary clarifications provided. Therefore, I consider the article ready for publication.

Reviewer 4 Report

Comments and Suggestions for Authors

The author has already answered or modified the doubts or questions I raised.